# VidMan: Exploiting Implicit Dynamics from Video Diffusion Model for Effective Robot Manipulation

**Youpeng Wen**[1][*], **Junfan Lin**[2][*], **Yi Zhu**[3], **Jianhua Han**[3],
**Hang Xu**[3], **Shen Zhao**[1][†], **Xiaodan Liang**[12][†]
[1]Shenzhen Campus of Sun Yat-Sen University,
[2]Peng Cheng Laboratory, [3]Huawei Noah's Ark Lab

## Abstract

Recent advancements utilizing large-scale video data for learning video generation models demonstrate significant potential in understanding complex physical dynamics. It suggests the feasibility of leveraging diverse robot trajectory data to develop a unified, dynamics-aware model to enhance robot manipulation. However, given the relatively small amount of available robot data, directly fitting data without considering the relationship between visual observations and actions could lead to suboptimal data utilization. To this end, we propose **VidMan** (**Vid**eo Diffusion for Robot **Man**ipulation), a novel framework that employs a two-stage training mechanism inspired by dual-process theory from neuroscience to enhance stability and improve data utilization efficiency. Specifically, in the first stage, VidMan is pre-trained on the Open X-Embodiment dataset (OXE) for predicting future visual trajectories in a video denoising diffusion manner, enabling the model to develop a long horizontal awareness of the environment's dynamics. In the second stage, a flexible yet effective layer-wise self-attention adapter is introduced to transform VidMan into an efficient inverse dynamics model that predicts action modulated by the implicit dynamics knowledge via parameter sharing. Our VidMan framework outperforms state-of-the-art baseline model GR-1 on the CALVIN benchmark, achieving a 11.7% relative improvement, and demonstrates over 9% precision gains on the OXE small-scale dataset. These results provide compelling evidence that world models can significantly enhance the precision of robot action prediction. Codes and models will be public.

## 1 Introduction

In the rapidly advancing field of robotics, accurately predicting and executing precise actions based on sensory inputs is crucial. While traditional approaches [1–5] for robot manipulation often rely on labor-intensive hand-engineered features and models prone to errors, data-driven methods [6–8] offer promising solutions. However, the challenge lies in the difficulty and cost of acquiring high-quality robotic data. Recent advancements [9–12], particularly those utilizing large-scale online video data for learning a video generator, demonstrate significant potential in comprehending complex physical dynamics of the real world. These models, trained on diverse datasets [13, 8], possess a nuanced understanding of the world, suggesting the feasibility of amalgamating and leveraging varied robot visual trajectory data [14–16] to develop a unified dynamics-aware model for enhanced robot manipulation. Yet, achieving this unification poses challenges; merely fitting data without considering the relationship between visual observations and actions could lead to suboptimal utilization of the data. Hence, there is a pressing need to develop efficient training mechanism and model architecture

---

[*]Equal contribution, [†]Corresponding author

38th Conference on Neural Information Processing Systems (NeurIPS 2024).

Figure 1: VidMan's two-stage training paradigm mirrors dual process theory: its first stage (like System 2) pre-trains on understanding environment dynamics through video diffusion, forming a foundation for accurate action prediction, while its second stage (like System 1) was adapted from the first stage to leverage the learned dynamics knowledge for rapid, low-level action inference.

that can effectively leverage existing cross-robot and cross-scene data to enhance action prediction accuracy.

As shown in Fig. 1, to optimize the utilization of diverse robot data [8], we draw upon insights from neuroscience's dual process theory [17–19], which unveils the complex mechanisms of information processing and decision-making in the human brain. This theory distinguishes between two cognitive processes: System 1, responsible for rapid, intuitive responses based on immediate sensory inputs, and System 2, which involves slower, long-horizon planning grounded in abstract concepts and understanding of world dynamics [20]. Inspired by these insights, we adopt a two-stage paradigm for robot learning, exemplified by our innovative framework, terms as **VidMan** (**Vid**eo diffusion for robot **Man**ipulation). VidMan leverages the power of the video diffusion generation method Open-Sora [21] for robot imitation learning, tapping into the awareness of long-horizon dynamics inherent in video diffusion models to achieve more nuanced and dynamics-modulated robot action prediction. By learning different facets of data at distinct stages, VidMan acquires an inductive bias of inverse dynamics of robot control, wherein actions are the outcomes of state sequences, significantly enhancing the method's generalization performance, especially under scenarios with limited data.

Specifically, our VidMan employs a two-stage training mechanism, akin to the principles of dual process theory, to enhance stability and significantly improve data utilization: 1) In the first stage, namely the *Dynamics-aware Visionary Stage*, VidMan undergoes pre-training on the Open X-Embodiment [8] dataset (OXE) using a video diffusion model to predict future trajectories based on historical observations and language instructions. This stage involves the robot learning the dynamics of state transitions from data and accurately perceiving the current environmental state, enabling the model to develop a deep understanding of the environment's dynamics, forming a robust foundation for accurate action prediction; 2) In the second stage, dubbed the *Dynamics-modulated Action Stage*, VidMan incorporates a flexible yet powerful layer-wise self-attention adapter [22] to seamlessly integrate the pre-trained knowledge from the first stage into action prediction. Through shared neural architecture and parameters with the dynamics-aware visionary stage, this phase transforms VidMan into an implicit inverse dynamics model that infers dynamics-modulated actions without explicitly generating future visual trajectories, rendering it suitable for real-world robot control scenarios.

The performance of VidMan has been evaluated against SOTA baselines on the CALVIN [14] benchmark, where it achieves a 11.7% relative improvement. In addition, VidMan has shown notable effectiveness on the OXE small-scale dataset, achieving over 9% precision gains. This improvement is particularly evident when the data from the target robot is small, underscoring the effective data utilization of our method. Extensive ablation studies have been conducted to analyze the effects of various design decisions within our method. These experimental results suggest that VidMan represents a meaningful advancement in robotics, providing a valuable tool for developing more capable and responsive robotic systems.

## 2  Related Work

**Language-guided Robot Manipulation.** Language-guided robot manipulation has emerged as an elastic and straightforward approach to instructing robots to perform various tasks [14, 23]. Some existing methods [24, 25, 10, 26–30] leverage large language models (LLMs) to plan over the task domain and pass instructions to low-level action policies to generate robot actions. The hierarchical 2D policies [31–33] predict latent features or images of subgoals given a language instruction, which they feed into lower-level subgoal-conditioned policies. 3D policies [34, 35] combine 3D representations with diffusion objectives to learn manipulation from demonstrations using depth maps and camera extrinsics. Some methods [36, 37] also utilize 3D [38–40] or 2D [41–43] detection to identify objects and use constrained optimization methods to control robot operations Another line of work learns language-conditioned policies from unstructured play data, which contains demonstrations with and without language labels [44, 32]. These methods leverage sequence-to-sequence conditional variational auto-encoders to generate latent plans, which are then used to condition the action policy.

**Pre-training for Robot Manipulation.** The field of pre-training for robot learning has garnered significant attention in recent years [9, 26, 11, 45]. Some methods aim to learn useful visual representations through masked image modeling [9] and contrastive learning [11]. Previous research [46, 7, 47, 45, 25, 48] has focused on empowering robots and other agents with the capability to comprehend and execute language instructions, typically by learning policies conditioned on language. GR-1 [9] and RoboFlamingo [26] use a GPT-style framework to model action prediction as a token prediction task in CALVIN dataset [14], and achieve good results. Our approach is most similar to GR-1 in that it utilizes GPT-style framework to predict both video and action, while ours directly uses video's ability to capture future information to predict long sequences of actions.

## 3  Preliminaries

**Robot Dynamics and Inverse Dynamics.** The dynamics of a robot are typically characterized by the forward dynamics and the inverse dynamics. The forward dynamics describe how the current state and action determine the next state. Formally, given a state space $\mathcal{S}$ and an action space $\mathcal{A}$, the forward dynamics can be represented as the conditional probability distribution: $P(s_{t+1} \mid s_t, a_t)$, where $s_t, s_{t+1} \in \mathcal{S}$ are the robot state at timestep $t$ and $t+1$, respectively, and $a_t \in \mathcal{A}$ is the action taken. Conversely, inverse dynamics describe the probability of an action given a transition from one state to another. The inverse dynamics are particularly important in scenarios where only passive observation data is available, such as data collected from the internet, which often lacks explicit action information. Formally, the inverse dynamics are given by:

$$P(a \mid s_t, s_{t+1}). \tag{1}$$

**Observations to States.** In many real-world applications, direct access to the state space $\mathcal{S}$ is infeasible, and instead, sequences of image observations are provided. Let $\mathcal{O}$ denote the observation space, *e.g.*, image, proprio. To infer the underlying sequence of states $S = \{s_1, s_2, \ldots, s_T\}$ from a sequence of observations $O = \{o_1, o_2, \ldots, o_T\}$. This process can be formally described by finding the sequence of states that maximizes the posterior probability given the observations:

$$S^* = \arg \max_S P(S \mid O). \tag{2}$$

Using Bayes' theorem, this can be further expressed as:

$$S^* = \arg \max_S P(O \mid S)P(S), \tag{3}$$

where $P(O \mid S)$ is the likelihood of the observations given the states, and $P(S)$ is the prior probability of the sequence of states. This formulation is essential in various applications, including hidden Markov models (HMMs) and other state estimation techniques. With more observations, the likelihood $P(O \mid S)$ becomes more peaked around the true state sequence. This is because more data points allow for better discrimination between different state sequences.

## 4  Method

In this paper, we aim to enhance the precision of robot action prediction by exploiting the intricate dynamics encoded within robot visual trajectories. We propose VidMan, a novel framework leveraging

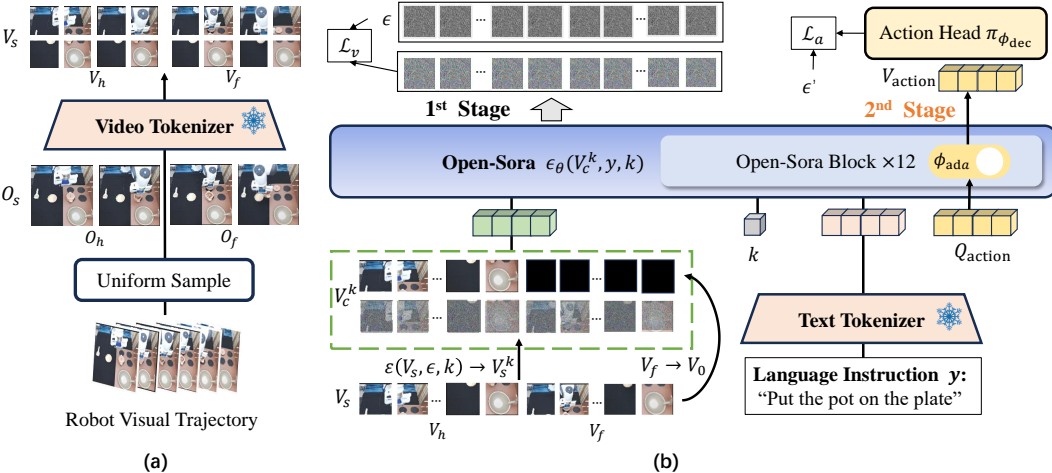

Figure 2: **Overview of VidMan.** (a) We use Video Tokenizer to tokenize the uniform sampled robot visual trajectory $O_s$ to video tokens $V_s$. (b) In the 1st Stage, we concatenate the video tokens processed through the diffusion process with the historical tokens along the channel dimension to form $V_c^k$. $V_c^k$ along with the language tokens and diffusion step $k$ are fed into Open-Sora for video prediction training. In the 2nd Stage, we use a learnable action token through a layer-wise adapter applied to the output of the Open-Sora Block to obtain tokens $V_{\text{action}}$ that integrate future frame information. $V_{\text{action}}$ are then fed into the Diffusion Action Head $\pi_{\phi_{dec}}$ for action prediction training.

Video diffusion for robot Manipulation. VidMan employs a dual-stage training strategy: in the first stage, the *Dynamics-aware Visionary Stage*, we enable the model to forecast and imagine potential future trajectories based on historical observations, leveraging the multi-frame prediction capability of the video diffusion model. Through this stage, the model is optimized to understand the dynamics of the environment. In the second stage, the *Dynamics-modulated Action Stage*, we introduce a lightweight layer-wise adapter to seamlessly integrate the visionary predictive stage with fast, adaptive action prediction. This approach decouples the knowledge of the world and embodiment into distinct processes while ensuring seamless integration through the training and utilization of shared parameters. Overview of our method is shown in Fig. 2. In the following sections, we will formulate each of these stages.

## 4.1 Dynamics-aware Visionary Stage

To endow the model with the knowledge of world dynamics, we formulate this knowledge acquisition stage as future image trajectory generation, which captures dynamic priors and predicts future state transitions more accurately according to Equ. (3). Specifically in our context, the goal is to predict future frames based on historical frames and language instructions. To achieve this, we leverage the capabilities of a multi-frame generation framework based on the video diffusion transformer model (VDT) Open-Sora [21], which has shown a remarkable ability to generate diverse and physically authentic successive frames aligned with language instructions. For simplicity, we use VDT to represent Open-Sora in the following content.

To prepare the training data, we utilize a pre-trained video tokenizer [49] to encode successive robot images in a trajectory $O_s = [O_h, O_f]$ into embeddings $V_s = [V_h, V_f]$, where $V_h \in \mathbb{R}^{m \times W \times H \times C}$ and $V_f \in \mathbb{R}^{n \times W \times H \times C}$ stand for embeddings for $m$ history images $O_h$ and $n$ future images $O_f$, respectively. During the forward diffusion process $V_s^k \leftarrow \varepsilon(V_s, \epsilon, k)$, the noise $\epsilon \sim \mathcal{N}(0; 1)$ is added to trajectory embeddings $V_s$ according to the diffusion step $k \in [1, K]$. The diffusion model is optimized to predict the added noise $\epsilon$ from $V_s^k$ given the condition hints. As the condition hints, we pad $V_h$ with zero-valued embeddings $V^0$ to the same shape as $V_s^k$, which are concatenated with $V_s^k$ along the channel dimension to form the condition visual embeddings $V_c^k \in \mathbb{R}^{(m+n) \times W \times H \times 2C}$. And the language instruction $y$ is encoded by an text tokenizer [50] to obtain the language embedding. Mathematically, the denoised diffusion model is optimized according to:

$$\mathcal{L}_v(\theta) = \mathbb{E}_{(V_c^k, y, k)} \left[ \left\| \epsilon - \epsilon_\theta(V_c^k, y, k) \right\|_2^2 \right]. \tag{4}$$

In this training stage, we use only a third-person camera to predict the representation. This approach has two main advantages: a) most robot datasets include only third-person view data; b) training with a fixed third-person viewpoint can reduce the influence of view changes and help the model focus on predicting the transitions of the robotic arm itself. Additionally, our method can easily be extended to multiple cameras by simply inputting multiple cameras of viewpoint.

## 4.2 Dynamics-modulated Action Stage

According to Equ. (1), actions can be accurately predicted given the states with the inverse dynamics model. One straightforward approach to combine an inverse dynamics model with the VDT learned in the first stage is to separately construct an inverse dynamics model that maps from image observations to actions. During deployment, this model could predict actions from the generated observations of the dynamics-aware visionary stage. However, by this means, actions are only predictable after the VDT conducts a time-consuming iterative denoising diffusion process, which is not ideal for high-frequency robot control. Moreover, the accuracy of the actions is heavily dependent on the accuracy of the predicted observations. Since not all pixels are important for predicting actions, this method could introduce unnecessary bias and time costs. Additionally, learning a separate inverse dynamics model from scratch does not leverage the pre-trained parameters of the VDT.

To address these issues, we propose directly adapting the VDT into an inverse dynamics model. In this way, the dynamics knowledge and implicit states learned in VDT can be seamlessly leveraged to facilitate the prediction of actions. Below, we introduce a lightweight adapter module that effectively transforms the VDT into an implicit inverse dynamics model. The output of this adapted model is then decoded into a sequence of actions using a diffusion-based action head.

**Implicit Inverse Dynamics Adapter.** To transform the VDT into an inverse dynamics model, we incorporated a layer-wise adapter which is inspired by [22] after each layer in VDT. Each adapter includes a multi-head self-attention and a feed-forward network (FFN) with a gated mechanism. We use $h$ learnable action tokens $Q_{\text{action}}$, concatenating them with the features ouput from each layer of the VDT, and input them into the layer-wise adapter. This fuses the knowledge of each layer of the VDT to produce $h$ final action embeddings $V_{\text{action}}$. Since we only reuse VDT parameters without its observation generation function, we disable the iterative denoising process by using a fixed diffusion step $k \leftarrow K$, which turns $V_s^k$ into pure Gaussian noise. Formally, the action embeddings are fused with the introduce layer-wise adapter parameterized by $\phi$:

$$V_{\text{action}} = \epsilon_{(\theta, \phi_{\text{ada}})}(V_c^K, y, K, Q_{\text{action}}), \tag{5}$$

where $\epsilon_{(\theta, \phi_{\text{ada}})}$ represents the VDT parameters incorporated with the layer-wise self-attention adapter with parameters $\phi_{\text{ada}}$. The flexibility of the layer-wise adapter allows for the integration of domain-specific knowledge into the action embedding. For example, if the downstream robotic manipulation task includes proprioceptive information, we can use a proprioception embedder to convert it into tokens, which are then concatenated with the output of each layer of the VDT to serve as the additional keys and values for the layer-wise adapter. Please refer to Appendix A.2 for more details.

**Diffusion-based Action Head.** The fused action embeddings, $V_{\text{action}}$, are subsequently translated into low-level control signals. To achieve this, we utilize a diffusion-based action head [51], $\pi_{\phi_{\text{dec}}}$, responsible for decoding action embeddings into executable action signals, such as determining the 7 degrees of freedom (DoF) for the end-effector pose and gripper status. Similar to the dynamics-aware visionary stage, the objective of action prediction is:

$$\mathcal{L}_a(\theta, \phi_{\text{ada}}, \phi_{\text{dec}}) = \mathbb{E}_{(V_{\text{action}}, l)} \left[ \left\| \epsilon' - \pi_{\phi_{\text{dec}}} \left( \varepsilon(V_{\text{action}}, \epsilon', l), l \right) \right\|_2^2 \right], \tag{6}$$

where $\epsilon' \sim \mathcal{N}(0, 1)$, $l \in [1, L]$ stands for the diffusion step and $\varepsilon$ denotes the forward diffusion process. Note that this diffusion-based action head is relatively small compared to the VDT, making its computational cost and time consumption almost negligible.

## 5 Experiment

In this section, we conduct comprehensive experiments to validate the effectiveness of our methods, along with in-depth ablation studies.

### 5.1 Experiment Settings

#### 5.1.1 2-stage Training Setting

**Settings of dynamics-aware visionary stage.** We initialized our model with the weights from Open-Sora's [21] 16x256x256 text-to-video model, which was trained on internet data [52–54]. Since we concatenated historical frames along the channel dimension, we append a zero matrix to the parameters of the tokens embedder layer in Open-Sora accordingly. We use Open X-Embodiment Dataset [8] to train our model, which was constructed by pooling 60 existing robot datasets from 34 robotic research labs around the world. We referred to the Octo [7] codebase to selected 25 datasets from it, which are heterogeneous not just in terms of the robot type, but also in the sensors (e.g., including or not including wrist cameras) and labels (e.g., including or not including language instrctions). Unless otherwise specified, we sampled video sequences from trajectories at intervals of 3, resulting in 4-frame video sequences, with 2 historical frames and 2 future frames. At this stage, we only used images from the third-person camera with $256 \times 256$ resolution. We use a maximum of $K = 1000$ diffusion step.

**Settings of dynamics-modulated action stage.** In this training stage, we used data from both the Open X-Embodiment (OXE) and CALVIN [14] datasets for training. We additionally incorporated images from a wrist camera as extra observations. For the data in OXE, we predicted 12 action steps, whereas for the CALVIN data, we predicted 10 steps. For the diffusion policy head, we set the noise addition steps to 100. We used a $224 \times 224$ third-person camera and a $224 \times 224$ wrist camera. In this stage, we set VDT's diffusion step to the maximum, i.e., $k = K$, and used pure noise as $V_s^K$. The VDT does not conduct the iterative denoising process. As for the diffusion-based action head, we set the maximum diffusion step as $L = 100$. More details can be found in Appendix A.2

#### 5.1.2 Benchmark and Baselines

**Simulation Evaluation:** The CALVIN benchmark utilizes the PyBullet[55] simulator and involves a Franka Panda Robot arm interacting with various environments labeled A, B, C, and D. Each environment includes a desk, a sliding door, a drawer, a button controlling an LED, a switch for a lightbulb, and three colored blocks (red, blue, and pink). These environments differ in desk textures and object positions. CALVIN provides 24 hours of unstructured tele-operated play data, with 1% annotated with language descriptions. Each instruction chain consists of five sequential language instructions for execution. Evaluation follows a zero-shot generalization setup, training models on environments A, B, and C and testing on D. Performance metrics include success rates and average completion of sequential tasks, as per prior studies[26, 9]. Notably, CALVIN lacks a motion planner, requiring all models to predict robot pose trajectories.

**Offline Evaluation:** We also report offline metrics, including the average of xyz accuracy and euler angle accuracy (Avg xyz ang) and MSE for end-to-end action prediction on Bridge [56], Taco Play [57], Cable Routing [58] and Autolab UR5 [59], which are presented in OXE [8]. Following Octo [7], we use continuous action space. XYZ accuracy measures the precision of the robot's predicted 3D position (X, Y, Z coordinates) compared to the ground truth values during evaluation. Euler angle accuracy measures the precision of the robot's predicted orientation angles (rotations around X, Y, and Z axes) compared to the ground truth values during offline evaluation. Specifically, XYZ accuracy refers to whether we predicted the XYZ delta within 0.5 radians and 50% of the norm while in motion. Euler angle accuracy indicates whether we predicted the rotation delta within 0.5 radians during movement. Additionally, we reported the mean squared error (MSE) which reflects how well each model predicts the actions.

**Baselines:** On the CALVIN benchmark, we compare our approach to the hierarchical 2D policies of MCIL [31], HULC [32], and SuSIE [33], which predict latent features or subgoal images based on language instructions and feed them into lower-level subgoal-conditioned policies. These methods can train the low-level policy using the full CALVIN dataset, rather than being restricted to the language-annotated subset. We also compare against large-scale 2D transformer-based policies like RT-1 [47], RoboFlamingo [26], and GR-1 [9], which pretrain on extensive interaction or observational (video-only) data. Additionally, to better highlight the performance of our method, we compare with the 3D methods, 3D Diffusion Policy [34] and 3D Diffuser Actor [35]. Both share a similar goal of combining 3D representations with diffusion objectives to learn manipulation from demonstrations using depth maps and camera extrinsics. As for OXE benchmark, we compared our method with

RT-1-X [8] and Octo [7]. RT-1-X is an openly available generalist robot policy which are trained on OXE using RT-1 [47] framework. Octo is a transformer-based diffusion policy that supports flexible task and observation definitions, pretrained on OXE. These baselines are used to validate the benefit of our two-stage training strategy since both RT-1-x and Octo are optimized with all data in a train-from-scratch manner. To demonstrate the effectiveness of using diffusion models in learning dynamics priors and modeling states in the first stage, we implemented a GPT-style (VidMan-GPT) policy, as a baseline. This baseline autoregressively predicts the next images and actions in the first stage. This allows us to compare, in a fair manner, the benefits of simultaneously predicting multiple frames in a diffusion-like manner versus predicting only the next frame in a GPT-like fashion for modeling dynamics priors and state representations. We only predict a single frame and the corresponding action. We discarded the diffusion scheduler and used random masking [60] to reconstruct images to calculate the video loss.

## 5.2 Comparison Results

Table 1: **Zero-shot long-horizon evaluation on CALVIN.** *All* denotes that the model is trained on the entire dataset, including visual data without language annotations, while *Lang* refers to training on only the language-labeled data. Our method outperforms the hierarchical 2D policies (MCIL [31], HULC [32] and SuSIE [33]) and large-scale 2D transformer-based policies (RT-1 [47] RoboFlamingo [26] and GR-1 [9]), while also remaining competitive compared to 3D-based policies (3D Diffusion Policy [34] and 3D Diffuser Actor [35]).

| Method | Training Data | Tasks completed in a row | | | | | |
|---|---|---|---|---|---|---|---|
| | | 1 | 2 | 3 | 4 | 5 | Avg. Len. |
| 3D Diffusion Policy [34] | Lang | 28.7 | 2.7 | 0 | 0 | 0 | 0.31 |
| MCIL [31] | All | 30.4 | 1.3 | 0.2 | 0 | 0 | 0.31 |
| HULC [32] | All | 41.8 | 16.5 | 5.7 | 1.9 | 1.1 | 0.67 |
| RT-1 [47] | Lang | 53.3 | 22.2 | 9.4 | 3.8 | 1.3 | 0.9 |
| RoboFlamingo [26] | Lang | 82.4 | 61.9 | 46.6 | 33.1 | 23.5 | 2.48 |
| SuSIE [33] | All | 87 | 69 | 49 | 38 | 26 | 2.69 |
| GR-1 [9] | Lang | 85.4 | 71.2 | 59.6 | 49.7 | 40.1 | 3.06 |
| 3D Diffuser Actor [35] | Lang | 93.8 | 80.3 | 66.2 | 53.3 | 41.2 | 3.35 |
| VidMan (Ours) | Lang | 91.5 | 76.4 | 68.2 | 59.2 | 46.7 | 3.42 |

**Multi-Task Performance.** We first conducted the initial stage of training on OXE, then utilized CALVIN's static camera and wrist camera as the two input sources for the second stage of training on CALVIN. Following the setup in GR-1, we used the portion of the CALVIN dataset that includes language instruction labels, while also incorporating CALVIN's proprioceptive state data as additional input into the layer-wise adapter. Tab. 1 compares the performance of mainstream methods such as SuSIE, RT-1, RoboFlamingo, GR-1, and 3D Diffuser Actor against VidMan. Compared to hierarchical 2D policies like SuSIE, which struggle to effectively leverage language information at the low level despite being able to train on the entire dataset ("All"), these methods generally underperform in terms of generalization compared to end-to-end policies like GR-1 and RoboFlamingo. Our method adopts an end-to-end training approach, outperforming the best hierarchical architecture, SuSIE, by 0.73 in Avg. Len., while using less data. When compared to large-scale 2D transformer-based policies such as GR-1 and RoboFlamingo, which also use an end-to-end output approach, our method is highly comparable. Thanks to pretraining on the large-scale robotics dataset OXE and utilizing the Open-Sora architecture, which is more vision-friendly, our method outperforms GR-1 by a clear margin on average length (11.7% relative improvement). In comparison to 3D Diffuser Actor, which leverages depth information to predict actions, our model remains competitive due to the ability to pretrain on large-scale datasets without depth information, which are easier to collect.

**Offline Performance.** To evaluate offline performance, in this part, the second stage of our model is optimized with OXE dataset. We evaluate performance on in-distribution tasks. Specifically, we conduct two parts of evaluation: 1) the evaluation on domains that have small-scale sub-datasets in OXE (Taco Play, Cable Routing and AUTOLab UR5), where we would expect pretraining with larger datasets can significantly improve performance; 2) the evaluation on domains with large-scale dataset (Bridge), where we expect further improvement to be more challenging. We compare our results with RT-1-X, Octo-small, Octo-base, and VidMan-GPT on the four sub-datasets. We compared the average

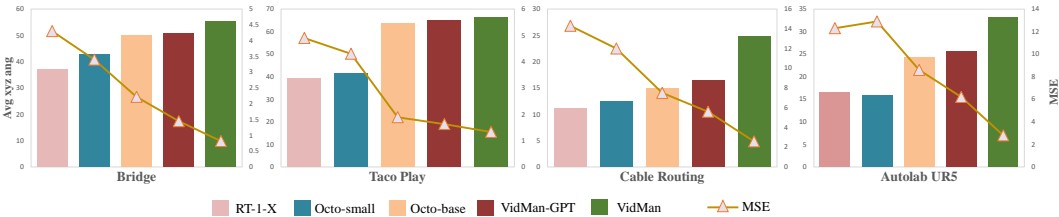

Figure 3: **Offline Performance.** The average accuracy (Avg xyz ang) of xyz accuracy and angle accuracy and MSE correspond to the left and right y-axes of the graph respectively. All models were trained on OXE and validated on offline performance across four datasets. VidMan outperformed Octo-base [7] by 5.6% on Bridge, 2.6% on Taco Play, 9.9% on Cable Routing, and 9.0% on Autolab UR5. Additionally, Our method also shows improvements over the VidMan-GPT approach.

Table 2: **Ablation Studies on Key Factors of VidMan..** We conduct finetune experiments on CALVIN. Average length is used. Best practice settings are marked in gray .

(a) **2-stage training.** Training only the action prediction in the second stage is crucial.

| Setting | CALVIN |
|---|---|
| co-train | 2.70 |
| action-only | 3.42 |

(b) **Pretrain type.** Video pretraining on domain-specific data is important.

| Type | CALVIN |
|---|---|
| w/o Pre-train | 2.89 |
| w/ Ego4d | 3.29 |
| w/ OXE | 3.42 |

(c) **Effect of layer-wise adapter.** Layer-wise adapter is important for performance improvements.

| Setting | CALVIN |
|---|---|
| w/o adapter | 1.54 |
| freeze | 2.98 |
| no freeze | 3.42 |

values of xyz accuracy and angular accuracy (Avg xyz ang), the higher the better, as well as the MSE (the lower the better). As shown in Fig. 3, it can be observed that our method outperforms the state-of-the-art open-source method, Octo, in both evaluation settings. Particularly on the small-scale sub-datasets CableRouting and Autolab UR5, our method improves the offline average xyz angle accuracy by 9.9% and 9.0% over Octo, respectively. The key difference between Octo and our method is in the optimization approach. Octo employs a co-training strategy, utilizing all data at once. However, our two-stage training strategy demonstrates a significant improvement, especially in domains with limited data. This improvement is attributed to our approach's ability to utilize the training data more efficiently by implicitly leveraging the inductive bias of inverse dynamics. Regarding Taco Play, the improvement we achieved is less notable. This is likely because the Taco Play scenes are relatively simple, and various methods tend to achieve similar performance levels in such environments. Additionally, our method shows improvements of 5.6% on the large-scale dataset Bridge which also demonstrates the effectiveness of our method. We have also achieved significant improvements over our own baseline, VidMan-GPT, across all four datasets, demonstrating the superiority of our model architecture. Details can be found in Tab. 7

### 5.3 Extra Ablation Studies

In this section, we conduct ablation studies for VidMan to answer the following questions: 1) Can co-training obtain a similar performance to our two-stage training? 2) Whether pretrained with general video data on the internet help improve the performance? 3) How much performance is gained by using the layer-wise self-attention block? 4) Does increasing the frame sampling interval positively impact action prediction? In the following, we answer each of these questions.

**The importance of two-stage training.** In our second stage, we only use action as a supervision signal to learn an implicit inverse dynamics model $P(a_t \mid s_t, s_{t+1})$ in Equ. (1). In other words, it suggests that we should not co-train with both actions and frames generation at the same time, otherwise a joint distribution $P(a_t, s_{t+1} \mid s_t)$ is learning. To validate this viewpoint, we conducted experiments where the frames generation loss $\mathcal{L}_v$ is also used in the second stage along with $\mathcal{L}_a$. The result of this variant is presented as "co-train" in Tab. 2a. In comparison to using only $\mathcal{L}_a$ in the second stage, namely "action-only" in the table, the average score of "co-train" degrades severely and approaches the performance of baseline methods in Tab. 2a. This further supports the effectiveness of our two-stage training strategy.

Table 3: **Effect of Frame Sampling Interval.** Setting the frame sampling interval to 3 is effective.

| interval | FID↓ | FVD↓ | Bridge Dataset | | | CALVIN |
| | | | MSE ↓ | xyz ↑ | angle ↑ | Avg. Len. ↑ |
|---|---|---|---|---|---|---|
| 1 | 29.5 | 327 | 2.80 | 42.8 | 46.6 | 2.24 |
| 2 | 32.1 | 345 | 1.29 | 48.1 | 51.2 | 2.82 |
| 3 | 38.4 | 376 | **0.89** | **51.5** | **58.2** | **3.42** |
| 4 | 51.9 | 422 | 1.20 | 48.2 | 53.1 | 3.03 |

**Pretrained with general video data.** To evaluate whether pretraining with general video data (not specific to robotics) from the internet helps improve performance, we conducted two control experiments. "w/o Pre-train" refers to our model without video pretraining, "w/ Ego4d" represents pretraining during the first stage using the general dataset Ego4d [13] for video prediction, and "w/ OXE" refers to pretraining using the robot-specific dataset OXE. For the Ego4d dataset, we followed the same processing pipeline as GR-1. As shown in Tab. 2b, pretraining with robot-specific video data resulted in a 0.53 increase in the average task length on CALVIN compared to "w/o Pre-train", while pretraining with general data ("w/ Ego4d") slightly decreased performance by 0.13. This suggests that training on web-based general data can yield marginal improvements, and pretraining in robotics domains significantly boosts performance.

**Effect of layer-wise adapter.** To measure the effect of the layer-wise adapter, we removed it (w/o adapter) and directly extracted the observed information into the action policy head by concatenating learnable action token to observation tokens before Open-Sora blocks. We found that this significantly reduced performance, comparing row 1 and row 3 in Tab. 2c. Additionally, we found that training only the layer-wise adapter while keeping the Open-Sora blocks fixed leads to faster convergence, as shown in Fig. 4. The training loss of the "freeze" curve converges faster than the "not freeze" curve. If both the adapter and Open-Sora blocks are trained, better accuracy can be achieved, by comparing row 2 and row 3 in Tab. 2c. In this sense, we can quickly adapt to a specific robotic scenario with the adapter being trained. In brief, layer-

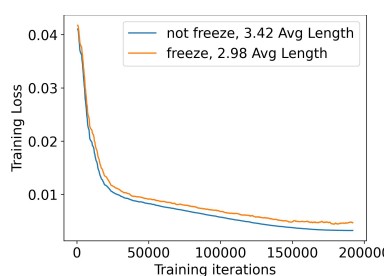

Figure 4: Efficiency comparison between two types of training.

wise adapter can provide good performance. If Open-Sora blocks' parameters are not fixed, better performance can be achieved; if the parameters are fixed, faster convergence can be obtained.

**Effect of frame sampling interval.** We compare the effectiveness of different frame sampling intervals (*i.e.* 1, 2, 3, and 4) on Bridge Dataset and CALVIN. With a larger frame interval, a longer historical information our model can perceive and a longer future information needed to be predicted. The total length of the four experimental frames is 4, including 2 historical frames and 2 future frames. Results are shown in Tab. 3. It is observed that increasing the intervals from 1 to 3 improves performance on Bridge and CALVIN. The potential reason is that consecutive frames are very similar, and predicting frames that are farther away from the current step helps the robot gain a better understanding of the future. However, when the interval increases from 3 to 4, this improvement saturates quickly. We speculate that predicting frames too far into the future may not offer effective guidance for immediate local action prediction.

## 6  Conclusion

In this paper, we propose VidMan, a novel framework utilizing video diffusion models for robot imitation learning, which addresses the limitations of current GPT-style paradigms in real-time applications. By combining a Dynamics-aware Visionary Stage, which develops a deep understanding of environment dynamics through pre-training on the Open X-Embodiment dataset, with a Dynamics-modulated Action Stage that efficiently integrates this knowledge into action prediction, VidMan achieves both high precision and computational efficiency. This two-stage approach, ensures robust and rapid action generation, significantly improving performance on benchmarks like CALVIN and the OXE dataset. In the future, we will expand VidMan to be able to perceive more dimensions of information.

**Acknowledgements** This work was supported in part by National Science and Technology Major Project (2020AAA0109704), National Science Foundation of China Grant No. 62476293, National Key Research and Development Program (Grant 2022YFE0112500), National Natural Science Foundation of China (Grants 62101607), Guangdong Outstanding Youth Fund (Grant No. 2021B1515020061), Shenzhen Science and Technology Program (Grant No. GJHZ20220913142600001), Nansha Key RD Program under Grant No.2022ZD014, The Major Key Project of PCL (No. PCL2024A04, No. PCL2023AS203).

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

# A  Appendix

The outline of the Appendix is as follows:

- Negative impacts and limitations.
- Details of the network and training process.
- Description and usage guide for OXE.
- Additional experimental results.
- Visualizations of video prediction and action prediction.

## A.1  Negative Impacts and Limitations

**Potential Negative Social Impact.** Our method has no ethical risk on dataset usage and privacy violation since all the benchmarks are publicly available and transparent.

**Limitations and Future Works.** Our current model operates solely on 2D vision and does not possess 3D perception capabilities. This limitation affects its performance on tasks that demand accurate 3D spatial understanding. Our future work will prioritize the integration of 3D perception into the model. Our model's capacity to comprehend complex human instructions is limited. It currently lacks the sophistication of a CLIP's language encoder. We are considering state-of-the-art Large Language Models (LLMs), such as LLama [61], as a promising avenue for future exploration to enhance these capabilities. The model's perception is not fine-grained. It processes the entire image as a single input for action prediction. We believe that incorporating fine-grained auxiliary inputs, such as object-bounding boxes or masks, could significantly enhance the model's performance. We are exploring ways to integrate such detailed inputs to refine the model's perceptual abilities.

## A.2  Network and Training Details

**Model details.** For the video autoencoder, we utilized Stabilityai's VideoAutoencoderKL [49] to encode the video into embeddings. For the text encoder, we employed CLIP's text encoder [62]. For the diffusion model, we used STDiT-XL/2 [21].

**Training details.** The video diffusion transformer of VidMan contains 12 layers (we reduced the original number of layers in Open-Sora to 12) and 16 heads with a hidden size of 1152. While training on OXE for video prediction in stage 1, we used 16 Nvidia V100 32GB GPUs, with a batch size of 24 per GPU resulting in a total batch size of 384. This stage will cost 42 hours. While training on OXE in stage 2, we used 16 Nvidia V100 32GB GPUs, with a batch size of 18 per GPU. We employed gradient accumulation, updating the parameters every 5 steps, resulting in a total batch size of 1440. This process took 32 hours. While training on CALVIN in stage 2, we used 8 Nvidia V100 32GB GPUs with a total batch size of 400. We use the AdamW [63] optimizer with an inverse square root decay learning rate schedule [64], with weight decay of 0.1. Training hyperparameters are shown in Tab. 4

Table 4: Training Hyperparameters

| Hyperparameters | 1st Stage | 2nd Stage OXE Training | 2nd Stage CALVIN Training |
|---|---|---|---|
| batch size | 384 | 1440 | 400 |
| learning rate | 2e-5 | 1e-4 | 1e-4 |
| dropout | 0.1 | 0.1 | 0.1 |
| optimizer | AdamW | AdamW | AdamW |
| weight decay | 0.1 | 0.1 | 0.1 |
| lr schedule | None | inverse square root decay | inverse square root decay |
| training steps | 100k | 300k | 20 epochs |

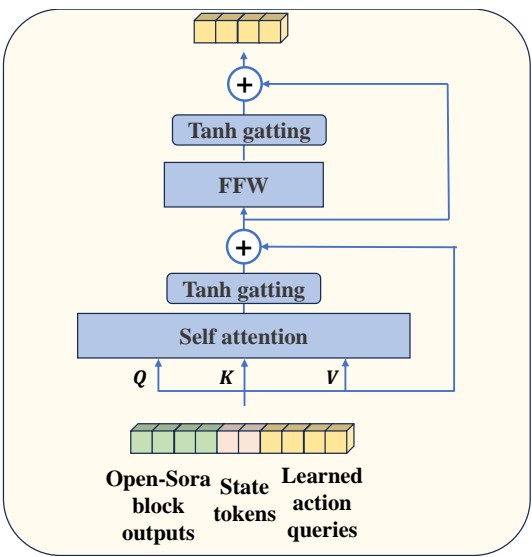

Figure 5: Our model utilizes a layer-wise adapter, which includes a self-attention layer and a feed-forward network (FFN). This block uses a gating mechanism to distill the information extracted by the Open-Sora block into the action query.

**Layer-wise adapter.** As shown in Fig. 5, following [22], our model employs a layer-wise self attention block, which comprises a self-attention layer and a feed-forward network (FFN). This block incorporates a gating mechanism to refine the information extracted by the VDT block into the action query.

## A.3   More Details of Open X-Embodiment Datasets

We trained VidMan using a curated selection from the Open X-Embodiment Dataset [8], which is a rich repository of robot learning datasets. Our training data encompasses a variety of robot forms, environments, and tasks, reflecting a broad spectrum of robot types, sensor configurations, and labeling styles, including the presence or absence of language instructions.

Inspired by the approach taken in Octo [7], we began by filtering out datasets that did not include image streams or those that did not utilize delta end-effector control. From the remaining datasets, we carefully selected those with the highest diversity and task relevance, omitting any that were too repetitive, had low-resolution images, or were too specialized.

We then divided the chosen datasets into two groups: "more diverse" and "less diverse", based on the variety of tasks and environments they represented. To ensure a balanced training, we assigned a higher weight to the "more diverse" datasets. Additionally, we adjusted the weight of some large datasets that had an abundance of data points to maintain a balanced mix.

To streamline the training process, we filled in any gaps in the camera channels with zero-padding and standardized the gripper action spaces across all datasets. This standardization meant that a gripper command of +1 consistently signified "the gripper is open", and 0 signified "the gripper is closed".

## A.4   Additional Experimental Results

**Why use pure noise in stage 2.** In the second stage, a key issue is how to utilize the video prediction model pre-trained in the first stage for action prediction. We replaced the noisy images from the first stage with zero embeddings, pure noise, and images without noise, and then concatenated them with historical frames along the channel dimension as input to the VDT. All experiments were trained and tested for offline validation on the Bridge dataset. These experiments used a small batch size of 192 and were trained for 100k iterations with VDT block frozen. The results are shown in Tab. 5. We found that using pure noise during training allows the model to achieve a certain level of performance during evaluation. One possible reason is that using pure noise aligns with the denoising process, effectively distilling the pre-trained model's future perception capabilities into action prediction.

Table 5: $V_c^k$ **type in stage 2.** Concatenating historical frames with noise in the channel dimension is an effective way.

| type | MSE↓ | xyz ↑ | angle ↑ |
|------|------|-------|---------|
| no_noise | 11.2 | 6.6 | 0.4 |
| pure_noise | **4.8** | **32.7** | **37.6** |
| pure_zero | 12.1 | 2.3 | 6.5 |

Table 6: **Effect of historical frames v.s. future frames.** The length of historical frames is more important than the length of future frames.

| history, future | FID↓ | FVD↓ | Bridge Dataset | | | CALVIN |
| | | | MSE↓ | xyz ↑ | angle ↑ | Avg. Len. ↑ |
|-----------------|------|------|------|-------|---------|-------------|
| m=1, n=1 | 28.9 | 318 | 4.20 | 37.0 | 41.6 | 2.09 |
| m=2, n=1 | 27.3 | 305 | 1.80 | 48.9 | 55 | 2.80 |
| m=2, n=2 | 38.4 | 376 | 0.89 | **51.5** | **58.2** | **3.42** |

**Effect of historical frames v.s. future frames.** To explore whether the length of historical frames or future frames has a greater impact on action prediction performance, we conducted analytical experiments with varying lengths of historical and future frames. We experimented with different settings: 1) history frame n=1, future frame m=1; 2) n=2, m=1; n=2, m=2. We kept the length of historical and future frames consistent across both the first and second training stages. The frame interval was set to 3. Results are shown in Tab. 6. Comparing row 1 and row 2, if the length of historical frames is shortened, there will be a significant decrease in performance on both Bridge and CALVIN. Comparing row 2 and row 3, shortening the length of future frames also leads to a decrease in performance, but not as significant as with the historical frames.

**Detailed numerical metrics on OXE data.** The numerical metrics in Fig. 3 are displayed in Tab. 7.

**Results on RLBench.** We also conducted experiments on RLBench [23]. We used 18 tasks from RLBench for multi-task robot learning evaluation. Each task includes 2-60 variations. For example, in the stack blocks task, "stack 2 red blocks" and "stack 4 purple blocks" are considered as two variants. During testing, our model has to handle novel object poses, randomly sampled goals, and randomly sampled scenes with different semantic instantiations of object colors, shapes, sizes, and categories. Each multi-task agent is evaluated independently on all of the tasks. Evaluations are scored either 0 for failures or 100 for complete successes. We report average success rates on 25 evaluation episodes per task for agents trained with 100 demonstrations. On the RLBench benchmark,

Table 7: **Detailed numerical metrics on OXE data.** All models were trained on OXE and validated on offline performance across four datasets. VidMan outperformed Octo-base [7] by 5.6% on Bridge, 2.6% on Taco Play, 9.9% on Cable Routing, and 9.0% on Autolab UR5. Additionally, Our method also shows improvements over the VidMan-GPT approach.

| Method | Bridge | | | | Taco Play | | | |
| | MSE↓ | xyz ↑ | angle ↑ | avg xyz ang ↑ | MSE↓ | xyz ↑ | angle ↑ | avg xyz ang ↑ |
|--------|------|-------|---------|---------------|------|-------|---------|---------------|
| RT-1-X | 4.3 | 36.2 | 38.1 | 37.2 | 4.9 | 36.4 | 42.1 | 39.3 |
| Octo-small | 3.4 | 40.3 | 45.5 | 42.9 | 4.3 | 39.0 | 44.1 | 41.6 |
| Octo-base | 2.2 | 45.8 | 53.8 | 49.8 | 1.9 | 62.5 | 65.0 | 63.8 |
| VidMan-GPT | 1.5 | 46.5 | 55.2 | 50.9 | 1.6 | 62.7 | 67.0 | 64.9 |
| VidMan | 0.8 | 52.3 | 58.4 | 55.4 | 1.0 | 64.3 | 68.4 | 66.4 |

| Method | Cable Routing | | | | Autolab UR5 | | | |
| | MSE↓ | xyz ↑ | angle ↑ | avg xyz ang ↑ | MSE↓ | xyz ↑ | angle ↑ | avg xyz ang ↑ |
|--------|------|-------|---------|---------------|------|-------|---------|---------------|
| RT-1-X | 14.3 | 8.7 | 13.5 | 11.1 | 12.3 | 13.9 | 19.2 | 16.6 |
| octo-small | 12.0 | 10.0 | 15.0 | 12.5 | 12.9 | 13.8 | 18.0 | 15.9 |
| octo-base | 7.5 | 12.5 | 17.2 | 14.9 | 8.6 | 23.2 | 25.3 | 24.3 |
| VidMan-GPT | 5.6 | 13.2 | 19.8 | 16.5 | 6.2 | 24.3 | 27.0 | 25.7 |
| VidMan | 3.3 | 22.4 | 27.1 | 24.8 | 3.4 | 29.4 | 37.0 | 33.2 |

| Models | Open Drawer | Slide Block | Sweep to Dustpan | Meat off Grill | Turn Tap | Put in Drawer | Close Jar | Drag Stick | Stack Blocks | Screw Bulb |
|---|---|---|---|---|---|---|---|---|---|---|
| PerAct | 80 | 72 | 56 | 84 | 80 | 68 | 60 | 68 | 36 | 24 |
| RVT | 71.2 ± 6.9 | 81.6 ± 5.4 | 72.0 ± 0.0 | 88.0 ± 2.5 | 93.6 ± 4.1 | 88.0 ± 5.7 | 52 ± 2.5 | 99.2 ± 1.6 | 28.8 ± 3.9 | 48.0 ± 5.7 |
| Act3D | 93 | 93 | 92 | 94 | 94 | 90 | 92 | 92 | 12 | 47 |
| Ours | 94.4 ± 3.6 | 97.6 ± 4.4 | 92.8 ± 1.8 | 90.4 ± 2.2 | 96.8 ± 3.3 | 83.2 ± 1.8 | 88 ± 2.8 | 84.8 ± 3.5 | 48 ± 0.0 | 66.4 ± 2.1 |

| Models | Put in Safe | Place Wine | Put in Cupboard | Sort Shape | Push Buttons | Insert Peg | Stack Cups | Place Cups | Avg. Success | Inf. Speed |
|---|---|---|---|---|---|---|---|---|---|---|
| PerAct | 44 | 12 | 16 | 20 | 48 | 0 | 0 | 0 | 42.7 | - |
| RVT | 91.2 ± 3.0 | 91.0 ± 5.2 | 49.6 ± 3.2 | 36.0 ± 2.5 | 100.0 ± 0.0 | 11.2 ± 3.0 | 26.4 ± 8.2 | 4.0 ± 2.5 | 62.9 | 11.6 |
| Act3D | 95 | 80 | 51 | 8 | 99 | 27 | 9 | 3 | 65.1 | - |
| Ours | 67.2 ± 3.3 | 79.6 ± 0.0 | 32.8 ± 3.3 | 48.0 ± 0.0 | 89.6 ± 1.6 | 21.6 ± 3.6 | 18.2 ± 1.8 | 13.2 ± 1.8 | 67.4 | 18.3 |

Table 8: **Multi-Task Performance on RLBench-100.**

we compared our method with 3 methods: PerAct [65], RVT [66], Act3D [67]. To align with methods like RVT, we also reported the variance of 5 seeds in the evaluation on RLBench.

## A.5 Visualization

**Visualization of our video prediction.** We investigate the video prediction performance of VidMan trained on OXE in stage 1. Qualitative results are shown in Fig. 6. It can be observed that VidMan correctly predicts future frames, although some details (such as occluded objects) are missing. This predictive capability in the second stage can strongly guide action prediction as potential future forecasts.

**Visualization of our offline action prediction.** As shown in Fig. 7, we visualized VidMan's results on OXE offline evaluation. It can be observed that our model effectively fits the ground truth actions across various scenarios, potentially demonstrating its applicability to real robot tasks.

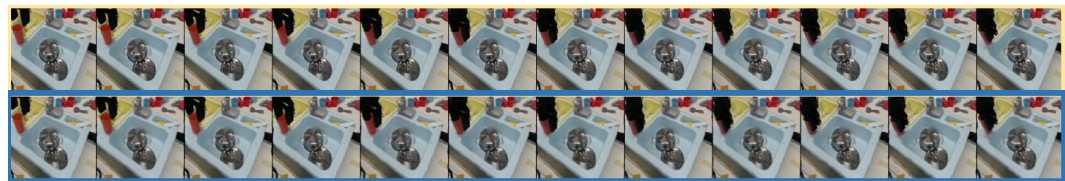

**"put cup from anywhere into sink"**

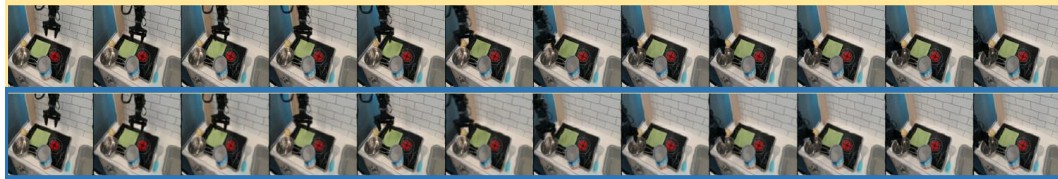

**"move the silver pot to the top right of the stove"**

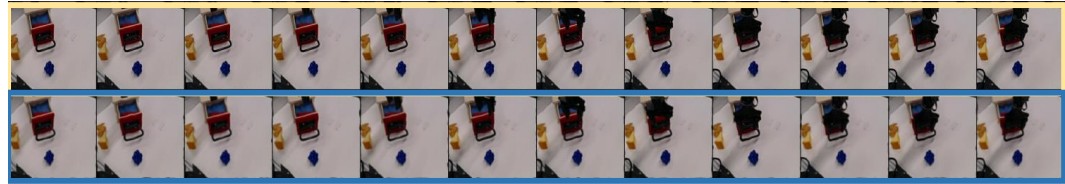

**"Close the draw"**

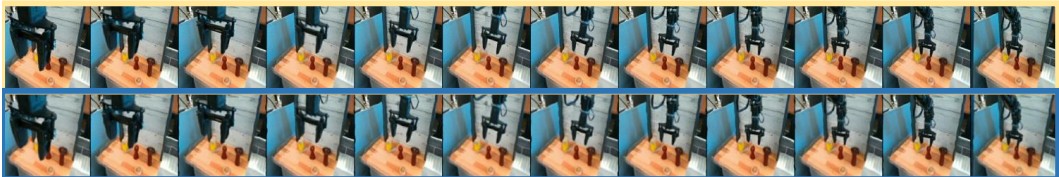

**"remove the cube from the tower and put it in the middle of the table"**

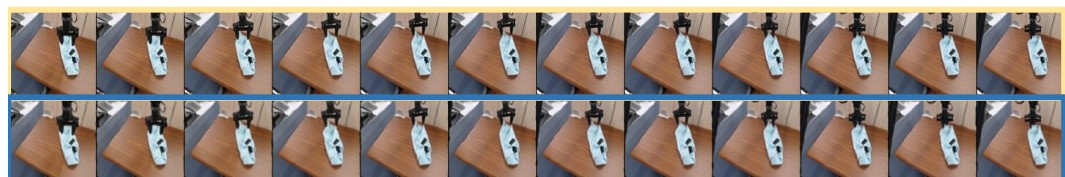

**"folding from left bottom to the right side"**

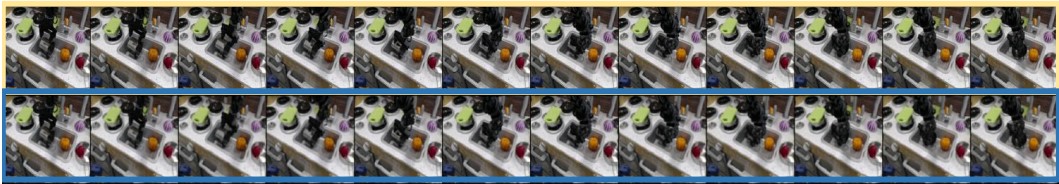

**"flip pot upright in sink"**

Figure 6: **Video prediction results on OXE.** The images in yellow boxes are ground-truth images; the images in blue boxes are predicted images. The language instruction is placed below the image.

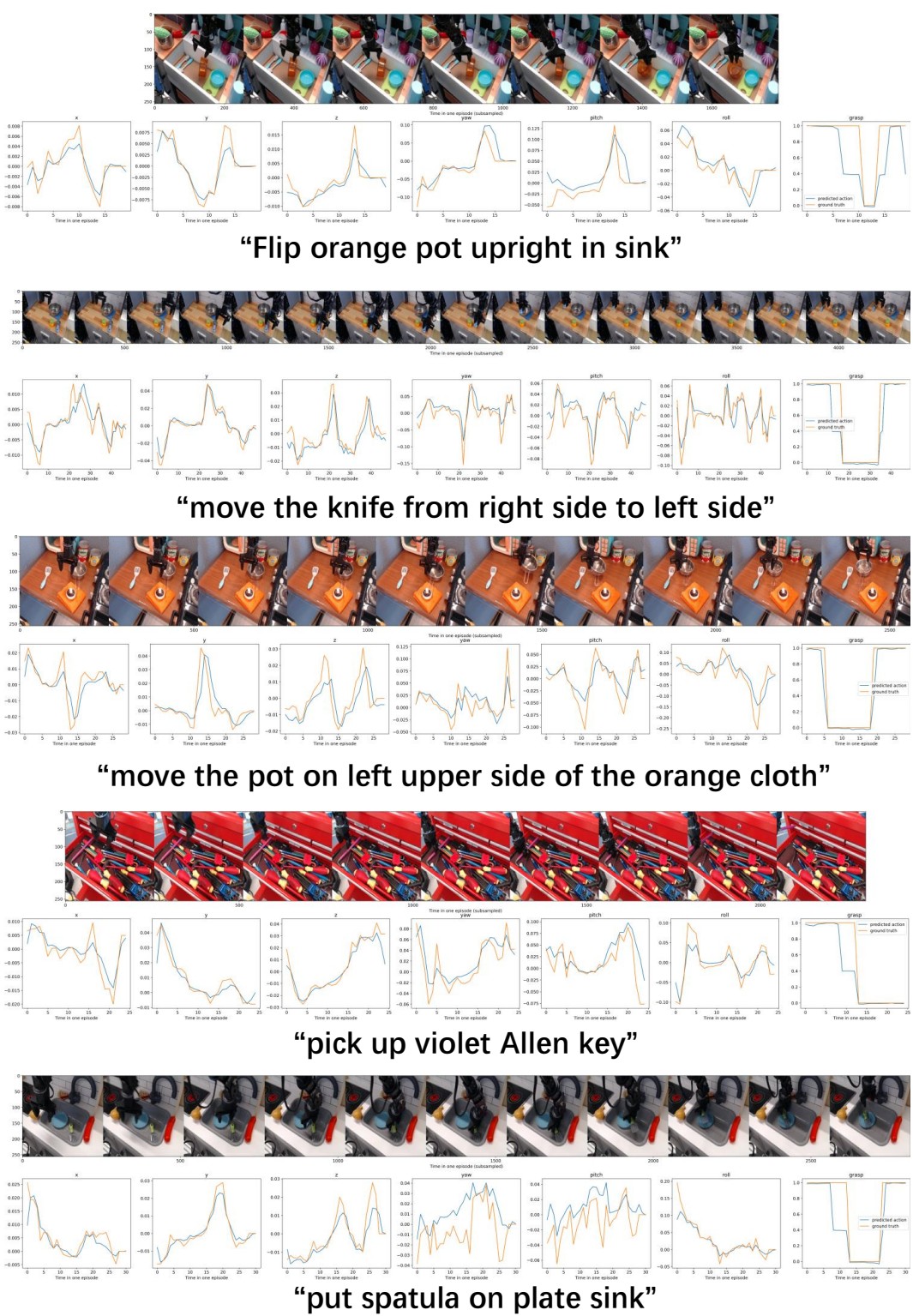

Figure 7: **Offline action prediction results on OXE.** The upper part of each group of images shows subsampled frames from an episode, while the lower part displays the true and predicted 7D pose results, including x, y, yaw, pitch, roll, and grasp over time.

