# OpenReview forum: "VidMan: Exploiting Implicit Dynamics from Video Diffusion Model for Effective Robot Manipulation"
_NeurIPS.cc/2024/Conference — NeurIPS 2024 poster_

### Official Review · Reviewer_Yu5P · 2024-06-26

**Soundness:** 1
**Presentation:** 3
**Contribution:** 2
**Rating:** 6
**Confidence:** 5

**Summary:**

The submission explores video pre-training for manipulation. Specifically, it trains a language-conditioned video prediction model to imagine future frames for the given task and an inverse dynamics diffusion model to predict actions, conditioned on the imagined future. The paper’s idea is that the first model can act as a slow process that thinks what will be done in the scene, while the second model is a fast action model, borrowing terminology from the dual process theory.

The experiments show multi-task manipulation results on RLBench and offline results on a subset of OXE datasets, while the paper also discusses interesting ablations.

**Strengths:**

1. The paper is clearly written and easy to follow in general.

2. The ablations show some interesting results, surprisingly negative but still valuable. Most notably, it seems that training the video model on web data doesn’t really add anything to the performance. This is good to know and it contrasts findings from recent papers such as [GR-1], which claim that video pre-training on non-robotic datasets can be helpful. Another interesting finding is that the second model better predicts actions only, rather than both actions and video frames. This also contrasts [GR-1].

3. As a technical contribution, using adapters to learn the inverse dynamics model on top of the weights of the video prediction model is an interesting idea.

References:

[GR-1] Unleashing large-scale video generative pre-training for visual robot manipulation

**Weaknesses:**

1. The major problem with this submission is the evaluation. Specifically, the results are very weak and unsafe for conclusions.

a. On RLBench, the paper seems to modify slightly the standard multi-task setup introduced in [PerAct] by changing the number of cameras used. Another possible difference is the number of seeds used. [PerAct] and other competitors [3DDA, Act3D, RVT, etc] use 5 seeds. I could not find this detail in the submission but I assume one seed was used, since there is no reported variance. Changing the setup doesn’t allow for direct comparison with prior works. As a result, the authors probably had to retrain the baselines.
While modifying the setup and retraining the baselines is not an issue per se, the selection of the baselines is highly disputable. [PerAct] is an old baseline published two years ago. Since then, [Act3D, RVT] achieved big leaps in 2023 and the current SOTA on RLBench is [3DDA]. Claiming SOTA on RLBench without comparing against actual SOTA models is wrong. In fact, with one camera only [3DDA] seems to achieve better results than the proposed approach.

b. Another controversy is the use of web data and OXE as pre-training. I mentioned in the strengths that revealing that web data don’t help is actually useful. The submission says “training on web data can help improve marginal performance” (line 313), which, while moderate, is actually an understatement; it’s better to say that it doesn’t really help. It would also be interesting to evaluate the usefulness of OXE pretraining; is training on RLBench enough for the model’s RLBench performance?
The above are important questions also since the baselines do not use any type of pre-training (save for the ablation VidMan-GPT). If the proposed approach uses many more data and still cannot compare against the baselines [3DDA, RVT, Act3D], then maybe this two-stage video pre-training is not the right direction. This would also be valuable to know.

c. The main competitor of this submission is [GR-1]: they also propose a two-stage video pre-training scheme, albeit with autoregressive models (instead of diffusion), trained on Ego4D (instead of OXE + web data) and training for both frame and action prediction at the second stage (instead of only actions). The submission acknowledges that [GR-1] is the most similar work (line 89). On that aspect, a fair and direct comparison against [GR-1] is required so that we know which design choices are indeed the best among the two, i.e., diffusion vs autoregressive, Ego4D vs OXE++, joint frame-action prediction vs action only. I appreciate the effort of putting together VidMan-GPT as a proxy for [GR-1] but it is not the same as the original model. Thus, it is unsafe to conclude among the two models. Instead, I would recommend that the submitted model is evaluated and ablated on CALVIN.

2. Another issue of the submission is its motivation. Most of the paper (for example the whole intro section) focuses on motivating the co-training with videos. But this motivation is somewhat generic, meaning that [GR-1] could also have the same motivation. Given that the submission acknowledges [GR-1] as closely related, what is the motivation beyond what supports [GR-1]? Borrowing from my previous paragraph, why do we need diffusion models for world modeling or any new video co-training model in general? What’s lacking from previous related approaches? Such targeted motivation would lead to stronger arguments for the proposed approach – if of course those can be validated through the experiments.

3. A last concern regarding motivation/presentation is the connection to dual process theory. From my knowledge about fast and slow thinking, the submission’s description of system 2 as “slow planning” is indeed fair. I can see why the manuscript argues that language-conditioned future prediction is a type of slow processing, since the agent judges about the changes it wants/expects to achieve and then executes. However, the description of system 2 in the related literature also refers to reasoning. I’m not exactly sure that planning through a black-box video diffusion model can really count as system 2 reasoning, since the model predicts pixels end-to-end and does not model the causal factors of variation a robot can affect, e.g. motion, pose difference. I would be reluctant to characterize the proposed approach as inspired by the dual process theory. I think it’s more inspired by representation learning and goal-conditioned policy learning [UniPi, SuSIE].



References:

[PerAct] Perceiver-actor: A multi-task transformer for robotic manipulation

[Act3D] Act3d: Infinite resolution action detection transformer for robotic manipulation

[RVT] Rvt: Robotic view transformer for 3d object manipulation

[3DDA] 3D Diffuser Actor: Policy Diffusion with 3D Scene Representations

[GR-1] Unleashing large-scale video generative pre-training for visual robot manipulation

[SuSIE] Zero-shot robotic manipulation with pretrained image-editing diffusion models

[UniPi] Learning Universal Policies via Text-Guided Video Generation

**Questions:**

Overall, the submission explores an interesting topic and I’d like to endorse the honesty of testing on RLBench, where 2D polices (like the proposed one) struggle, as well as not wasting time with real-world experiments. However, the experimental design is severely lacking, if not flawed. The baselines used are obsolete and the proposed method uses a lot of external data (which is good) but still fails to compete against strong baselines that are trained in-domain. In addition, a lot of discussion on and a direct comparison with [GR-1] is needed, as well as re-motivation based on these findings. The current version may lead to confusing or misleading conclusions due to lacking evidence. I encourage the addressing of all the issues.

To increase my score I'd like to see:

1. Direct comparison with GR-1 on the same dataset.

2. Discussion on the relation of this work to GR-1 and "Learning Universal Policies via Text-Guided Video Generation". The outcome of this should better motivate the approach, not as general two-step video-guided manipulation but on the importance of the specific contributions of the paper.

2. Inclusion of modern baselines on RLBench.


Minor: In total, the manuscript is easy to follow but there are grammatical or syntactical mistakes here and there, which could be easily fixed using online tools, if not careful proofreading.


--------------------------------------------
After the rebuttal, I'm increasing my score to weak accept.

**Limitations:**

Addressed

---

> ### Author Response · Authors · 2024-08-07
> **Response to Reviewer Yu5P [1/3]**
>
> Dear Reviewer,
>
> Thanks for your constructive comments. Below, we would like to address all the weaknesses and questions in details.
>
> **Q1:** `... by changing the number of cameras used ... the number of seeds used ...
> [ACT3D, RVT] achieved big leaps in 2023 and the current SOTA on RLBench is
> [3DDA]. `
>
> **A1:** We address this issue in the following four aspects.
> 1. We used an overhead camera and a wrist camera because the pre-training dataset OXE typically contains only these two cameras. To align the pre-training dataset with the fine-tuning dataset, we used a unified input. Since we adopted the STDiT structure of Open-Sora, it can easily be extended to multi-view input by simply concatenating tokenized image patches as input.
>
> 2. We thoroughly examined PerAct[1],  Act3D[2], RVT[3], 3DDA[4]. The original papers of PerAct and Act3D used 1 seed. RVT used 5 random seeds during inference because it did not directly output actions but used a sampling-based motion planner which introduced randomness. 3DDA used 3 random seeds but has not been formally published yet. To align with methods like RVT, we also reported the variance of 5 seeds in the evaluation on RLBench.
>
> 3. Directly comparing our method with RVT is not entirely fair. Our proposed method has the following key differences from RVT and other state-of-the-art approaches:
> a. RVT utilizes four RGB-D cameras, including a front camera, left shoulder camera, right shoulder camera, and wrist camera. In contrast, our method only uses an overhead camera and a wrist camera, and it does not incorporate depth information.
> b. RVT predicts the position and rotation angles of key frames, then uses a sampled motion planner to move the robotic arm to the corresponding position and rotate to the corresponding angle. Our method, on the other hand, outputs a 6-dimensional vector end-to-end, which is directly executed by the end-effector controller. An absolutely fair comparison may not be entirely feasible, but we can adopt some relatively fair comparisons. Specifically, we have standardized the first difference mentioned above by also using multiview input in our proposed method. Fortunately, due to the flexibility of our model, we only need to replace our overhead camera input with multiview cameras with the same resolution 128 x 128 (including front camera, left shoulder camera, right shoulder camera and wrist camera ) as the input of STDiT in the 2nd stage.
>
> 4. We compared our method with PerAct, RVT, and Act3D on RLBench-100 benchmark. We are not comparing with 3DDA for now, as it has not yet been published in a conference or journal temporarily. The results are as follows.
>
> | **Models**        | **opendrawer** | **slide block** | **sweep to dustpan** | **meat off grill** | **turn tap** | **put in drawer** | **close jar** | **drag stick** | **stack blocks** | **screw bulb** |
> |-------------------|----------------|-----------------|----------------------|--------------------|--------------|-------------------|---------------|----------------|------------------|----------------|
> | PerAct            | 80             | 72              | 56                   | 84                 | 80           | 68                | 60            | 68             | 36               | 24             |
> | RVT               | 71.2 ± 6.9     | 81.6 ± 5.4      | 72.0 ± 0.0           | 88.0 ± 2.5         | 93.6 ± 4.1   | 88.0 ± 5.7        | 52 ± 2.5      | 99.2 ± 1.6     | 28.8 ± 3.9       | 48.0 ± 5.7     |
> | Act3D             | 93             | 93              | 92                   | 94                 | 94           | 90                | 92            | 92             | 12               | 47             |
> | VidMan(multiview) | 94.4 ± 3.6 | 97.6 ± 4.4      | 92.8 ± 1.8           | 90.4 ± 2.2         | 96.8 ± 3.3   | 83.2 ± 1.8        | 88 ± 2.8      | 84.8 ± 3.5     | 48 ± 0.0         | 66.4 ± 2.1     |
> | **Models**        | **put in safe** | **place wine** | **put in cupboard** | **sort shape** | **push buttons** | **insert peg** | **stack cups** | **place cups** | **Avg score** | **Inf. Speed** |
> | PerAct            | 44              | 12             | 16                  | 20             | 48               | 0              | 0              | 0              | 42.7          | 4.9            |
> | RVT               | 91.2 ± 3.0      | 91.0 ± 5.2     | 49.6 ± 3.2          | 36.0 ± 2.5     | 100.0 ± 0.0      | 11.2 ± 3.0     | 26.4 ± 8.2     | 4.0 ± 2.5      | 62.9          | 11.6           |
> | Act3D             | 95              | 80             | 51                  | 8              | 99               | 27             | 9              | 3              | 65.1          | -              |
> | VidMan(multiview) | 67.2 ± 3.3      | 79.6 ± 0.0     | 32.8 ± 3.3          | 48.0 ± 0.0     | 89.6 ± 1.6       | 21.6 ± 3.6     | 18.2 ± 1.8     | 13.2 ± 1.8     | 67.4          | 18.3           |

---

> ### Author Response · Authors · 2024-08-07
> **Response to Reviewer Yu5P [2/3]**
>
> **Q2:** `Controversy of the use of web data and OXE as pre-training.`
>
> **A2:** We used web data and OXE for pre-training, emphasizing their role in situations with insufficient fine-tuning samples. Table 2 in our paper shows experiments conducted on RLBench with 10 demos, meaning each task has only 10 demonstrations for training. Regarding the controversy over using web data and OXE data, we have the following rebuttals.
> The reason why web data contributes very little is that it is non-optimal data. Open-Sora uses internet data, which has a significant gap with robotic manipulation scenarios. Another reason is that we only loaded the first 12 layers to make the model training more efficient. The OXE dataset or other large dataset indeed plays an important role in the generalization of the model. To clarify this point better, we have converted Table 2b into the table below. Additionally, we conducted experiments using other datasets for video prediction pre-training. Specifically, we replaced the first-stage pre-training dataset with Ego4D, while keeping other settings consistent with those in Table 2b. From the table below, we can see that our model's baseline performance on RLBench-10 is 51.2. When using robotic trajectory data (OXE), the performance improves to 57.1. When the pre-training dataset is non-robotic but still contains dynamics (Ego4D), the performance slightly decreases but still shows an improvement over the initial performance, reaching 55.5.
>
> | **Type**      | **RLBench-10** |
> |---------------|----------------|
> | w/ webdata    | 51.2           |
> | w/ Ego4D      | 55.5           |
> | w/ OXE        | 57.1           |
> | w/o web train | 56.3           |
>
>
> **Q3:** `Comparison of our results with GR-1.`
>
> **A3:** For a fair comparison with GR-1, we also conducted experiments with our model on the CALVIN benchmark, using the design choices of diffusion, OXE++, and action only. The results, as shown in following table, indicate that our model outperforms GR-1 on the same benchmark by 0.61.
>
> | **CALVIN** | **1** | **2** | **3** | **4** | **5** | **Avg.Len.** |
> |------------|-------|-------|-------|-------|-------|--------------|
> | GR-1       | 85.4  | 71.2  | 59.6  | 49.7  | 40.1  | 3.06         |
> | Ours       | 95.9  | 81.6  | 73.5  | 61.2  | 55.1  | 3.67         |
>
> Next, we discuss these three design choices.
>
> 1. diffusion vs autoregressive
>
> In Table 1 of our submission, we have already included a comparison between diffusion and autoregressive methods. Additionally, we evaluated the diffusion version of GR-1, as implemented in [5], on CALVIN. The following two tables show the results of two different designs of our method on RLBench-100 (copied from Table 1 of the submission) and the results of two different designs of GR-1 on CALVIN. The results indicate that, regardless of the method, diffusion performs better. This is likely because fine-grained diffusion design capture the intrinsic dynamics of videos better than coarse-grained autoregressive design.
>
> | **Model**           | **avg score** |
> |---------------------|---------------|
> | VidMan (autoregressive)     | 50.1          |
> | VidMan (diffusion)          | 66.4          |
>
> | **Model**   | **1** | **2** | **3** | **4** | **5** | **Avg.Len.** |
> |--------------|-------|-------|-------|-------|-------|--------------|
> | GR-1 (autoregressive)       | 85.4  | 71.2  | 59.6  | 49.7  | 40.1  | 3.06         |
> | GR-1 (diffusion) | 87.6  | 73.2  | 61.3  | 52.4  | 42.6  | 3.17         |
>
> 2.  Ego4D vs OXE
>
> Ego4D is a hand dataset containing approximately 3,500 hours of first-person perspective videos of human-object interactions. OXE++ is an expert-annotated robotic dataset with about 1.4M trajectory data points. Intuitively, in terms of performance, training on OXE is better than training on Ego4D. We conducted an ablation analysis of the pre-training data in the first stage for both benchmark, and the results are shown below. There is no doubt that using OXE for video prediction pre-training is better than using EGO4D. This is because OXE is expert-annotated robotic data with higher quality.
> Of course, Ego4D also has its advantages: it is easy to collect, allowing for data scale expansion and the resulting scale effects.
>
> | **Data** | **RLBench-10** |
> |----------|----------------|
> | w/ Ego4D | 55.5           |
> | w/ OXE   | 57.1           |
>
> | **Pretrained Dataset**         | **1** | **2** | **3** | **4** | **5** | **Avg.Len.** |
> |--------------------|-------|-------|-------|-------|-------|--------------|
> | Ego4D        | 88.7  | 77.5  | 63.8  | 54.1  | 45.3  | 3.294        |
> | OXE           | 95.9  | 81.6  | 73.5  | 61.2  | 55.1  | 3.673        |

---

> ### Author Response · Authors · 2024-08-07
> **Response to Reviewer Yu5P [3/3]**
>
> 3. joint frame-action prediction vs action only
>
> To investigate which is better, we conducted the following experiments: (a) our model on
> RLBench using joint frame-action prediction and action only, and (b) GR-1 on CALVIN
> using joint frame-action prediction and action only. The results indicate that different architectures may require different designs. Our model performs better when only predicting actions because we use cross-attention to integrate information into the action query and employ an additional diffusion head for output. Action prediction and frame prediction follow two separate branches, and frame prediction might disrupt the gradients back-propagated for action prediction. GR-1 performs better with joint prediction, possibly because it uses a single branch where frame prediction can better learn the representation of the video.
>
> | **RLBench-10**     | **Avg. Success** |
> |--------------------|----------------|
> | joint frame-action | 36.0           |
> | action only        | 57.1           |
>
> | **CALVIN**         | **1** | **2** | **3** | **4** | **5** | **Avg.Len.** |
> |--------------------|-------|-------|-------|-------|-------|--------------|
> | joint frame-action | 0.815 | 0.651 | 0.498 | 0.392 | 0.297 | 2.65         |
> | action only        | 0.823 | 0.609 | 0.425 | 0.318 | 0.225 | 2.4          |
>
> **Q4:** `A comparison of the research motivations between our method and GR-1.`
>
> **A4:** We can restate the differences of motivation from GR-1 in the following aspects:
> a. Diffusion-style methods, compared to GPT-style methods like GR-1, can predict longer future frames. This is because diffusion can simultaneously denoise multiple frames to generate future frames, while GR-1 can only generate one future frame image in a next-token generation manner based on historical frames. Generating future frames for more time steps has been proven beneficial, as can be seen in Table 7 of our paper. We copy the table below.
>
> | **Benchmark**          | **Bridge** |  **Dataset** | **RLBench-10** |
> |----------------|--------------------|--------------|---------------|
> | history,future | MSE                | xyz / angle | avg success     |
> | m=1,n=1        | 4.2                | 37    / 41.6  | 32            |
> | m=2,n=1        | 1.8                | 48.9  / 55    | 53.9          |
> | m=2,n=2        | 0.89               | 51.5  / 58.2  | 57.1          |
>
> b. GR-1 models inputs and outputs at the token level, which lacks perception of the image space and can lead to information loss. Our model, on the other hand, models inputs and outputs at the image level and has a spatial layer for fine-grained perception of the image.
>
> **Q5:** `The reasoning capability of diffusion model.`
>
> **A5:** We are not the first to utilize the reasoning capability of diffusion models. [9] propose an approach that leverages the expressiveness of latent diffusion to model in-support trajectory sequences as compressed latent skills. [10] use a latent diffusion model with a variational autoencoder that can reconstruct the neural network weights, to learn the distribution of a set of pretrained weights. [11] introduces diffusion models to search for topological logic. Additionally, OpenAI's official website [12] also describes Sora's capability to understand the physical world, e.g., “The model understands not only what the user has asked for in the prompt, but also how those things exist in the physical
> world.”
>
> **Q6:** `Discussion on the relation of our paper to "Learning Universal Policies via Text-Guided Video Generation"[7]`
>
> **A6:** [7] proposes explicitly generating future frames and using the variance between future frames and current observation images to generate action sequences. In contrast, we implicitly represent future frames without an additional denoising process, directly generating action sequences from the features. The benefit of this approach is that it eliminates the very time-consuming iterative denoising process, making it effective for real-time robotic manipulation.
>
> References：
>
> [1] Perceiver-actor: A multi-task transformer for robotic manipulation.
>
> [2] Act3d: Infinite resolution action detection transformer for robotic manipulation.
>
> [3] Rvt: Robotic view transformer for 3d object manipulation.
>
> [4] 3D Diffuser Actor: Policy Diffusion with 3D Scene Representations.
>
> [5] Unleashing large-scale video generative pre-training for visual robot manipulation.
>
> [6] Zero-shot robotic manipulation with pretrained image-editing diffusion models.
>
> [7] Learning Universal Policies via Text-Guided Video Generation.
>
> [8] https://github.com/EDiRobotics/GR1-Training.
>
> [9] Reasoning with Latent Diffusion in Offline Reinforcement Learning, ICLR 2024.
>
> [10] Diffusion-based Neural Network Weights Generation, ICML 2024.
>
> [11] Aligning Optimization Trajectories with Diffusion Models for Constrained Design Generation, NeurIPS 2023.
>
> [12] https://openai.com/index/sora/

---

> ### Comment · Reviewer_Yu5P · 2024-08-09
> **Great effort in rebuttal**
>
> I thank the authors for their clarifications and I endorse their effort on running several baselines, ablations and benchmarks. I feel the paper is much more complete now.
>
> My main concerns were regarding the evaluation and the motivation. On the evaluation aspect, I asked for a more direct comparison on RLBench and more ablations on the relationship with GR-1, preferably on CALVIN, which is a benchmark the authors hadn't used in the first version of the paper. The rebuttal did the following:
>
> * RLBench: direct comparison with RVT and previous SOTA. The results are obtained using 2D images only as input and no depth, please correct me if I'm wrong. This could be one of the best 2D models deployed on RLBench, which is dominated by 3D approaches. This suggests that the technical contributions of the paper are strong. However, I strongly recommend that 3DDA is included in the comparison, even if the proposed approach underperforms it. The paper will be very insightful if it shows where 2D policies have reached and where 3D policies stand. One suggestion would be to split the table into 2D and 3D. Another suggestion/question is whether the proposed method can incorporate RGB-D. Maybe the first stage can predict the future depth frames as well. It would be interesting to test that.
>
> * CALVIN: the newly added benchmark allows for the method to shine. Since GR-1's strong results, I expected that future prediction is important for this benchmark. An even broader question is whether video prediction is the best representation for forecasting in robotics, but I do recognize its strengths and it's impossible to answer this in one paper.
>
> * The ablations now are much clearer.
>
> Overall, the experimental updates significantly upgrade the paper's quality. Please consider including 3DDA in the comparison, as a reader working on the field I would find the clarity and candor very insightful.
>
> Regarding the motivation, I'm not completely satisfied, but I do recognize the rebuttal's effort in providing more context and discussion. I think the introduction could be punchier on the specific contributions of the paper that made this idea work, rather than promoting the idea in general, because it's not novel on its own. I feel that this paper has more technical contributions that theoretical and it's fair to be portrayed as such. I'm also not a fan of connecting the two-stage approach to dual process theory, it feels a bit forced.
>
> I disagree with 5fDE calling the contributions of this paper "engineering", but I believe they convey what I wrote above, that this work has solid technical contributions, rather than theoretical. I do agree with 5fDE that NeurIPS papers should have deep insights and inspire future research. I believe that this papers provides some insights, even if the scope is limited within the field of imitation learning for tabletop manipulation. Since i) NeurIPS is not mainly a robotics conference and ii) the paper stands more on the technical/experimental side rather than theoretical, I feel that these contributions are good enough for acceptance, but I strongly disagree with yiak proposing this paper for award and I honestly advise the AC not to recommend this paper for an award.
>
> Therefore, I'm increasing my score to borderline accept. However, I would appreciate complete candor in the final version which should include both CALVIN and RLBench (together with the other sets used), as well the results of all related approaches, even if they outperform the proposed approach - especially since this happens on RLBench, which is 3D-dominated, as I mentioned earlier. Thank to the authors for the discussion.

---

> > ### Author Response · Authors · 2024-08-09
> > **Thank you for your suggestion and affirmation**
> >
> > Dear Reviewer,
> >
> > Thank you for your thorough review and for taking the time to provide detailed feedback on our work. We are pleased to hear that the additional baselines, ablations, and benchmarks have significantly improved the completeness of our paper.
> >
> > - RLBench: We appreciate your recognition of our model's performance using only 2D images. Indeed, our results are obtained without using depth information, and we agree that this highlights the strong technical contribution of our approach to further exploit the upper bound with 2D sensory inputs. And we appreciate your suggestion to include a comparison with 3DDA. We understand the importance of demonstrating how our 2D approach compares with 3D policies, and we will incorporate this comparison in the final version of the paper. Additionally, we are intrigued by your suggestion to explore the incorporation of RGB-D data and the prediction of future depth frames in the first stage of our model. We will investigate this direction in future work as it aligns with our goal of broadening the applicability and robustness of our approach.
> >
> > - CALVIN: We are glad that the inclusion of the CALVIN benchmark has addressed your concerns regarding the relationship with GR-1 and demonstrated the strengths of our method in this setting. We appreciate your broader question in pursuit of a more comprehensive representation of forecasting in robotics and we have faith that our work has provided a step forward in exploring this avenue and its potential applications.
> >
> > Thank you for your feedback regarding the motivation and the framing of our contributions. We will revise our description to be more conservative when referencing the dual process theory, especially where the connection may not be as strong. We will also place greater emphasis on showcasing the significant technical contributions and the positive impact they have on the field, as you've pointed out.
> >
> > We take your suggestion for candor in the final version seriously and will ensure that all related approaches, including those that outperform our method, are included in the final results. We believe that this transparency is crucial for advancing the field and providing readers with a clear understanding of the current state of the art.
> >
> > We appreciate your constructive feedback and are grateful for your increased score. Your suggestions have been invaluable in refining our work.

---

> > > ### Comment · Reviewer_Yu5P · 2024-08-10
> > > **Score increase**
> > >
> > > After reading all the responses again, I believe that if the authors indeed revise the paper as suggested in their latest comment above, it will be a stronger submission. To this end, I'm updating my score from 5 to 6 (originally I had scored this paper as 2, as I believe that the original submission had many flaws).

---

> > > > ### Author Response · Authors · 2024-08-11
> > > > **Response to Reviewer Feedback on Planned Revisions**
> > > >
> > > > Thank you very much for your thorough review and for your constructive feedback throughout the review process.  We are pleased to hear that you recognize the improvements in our paper and have increased the score accordingly.
> > > >
> > > > We have carefully considered all of your suggestions and are committed to revising the paper as you recommended.  The planned revisions will further strengthen our submission, and we are grateful for your insights that have guided us in this direction.

---

### Official Review · Reviewer_5fDE · 2024-07-08

**Soundness:** 2
**Presentation:** 3
**Contribution:** 2
**Rating:** 5
**Confidence:** 4

**Summary:**

This paper introduces a novel framework that utilizes a pre-trained video diffusion model to learn dynamic information and then adapt it to output actions for manipulation tasks by incorporating an action-aware adapter. In the second stage, this framework can transfer the learned dynamic knowledge for action learning in a single denoising step.

**Strengths:**

The idea presented in this paper, which aims to enhance decision-making through video pre-training, is not new in the fields of RL and robotics. However, I believe this topic is compelling and this paper could inspire further research in this direction.
- This paper is well-written, and the presentation is clear.
- This paper incorporates a layer-wise adapter to utilize the pre-trained 1-stage parameters for action learning, which is common in VLM but novel in the robotics area.
- Success rates on 15 tasks in RLBench and prediction accuracies on OXE datasets demonstrate the efficacy of the proposed method

**Weaknesses:**

- Although the proposed adapter is validated to be effective in the experiments, this paper lacks more evidence to demonstrate its superior performance compared with other approaches. For example, can we use a projector after the open-sora outputs instead of layer-wise adapters? Can we utilize the hidden representations of predicted videos like [1]?
- While I appreciate the authors' efforts in proposing a new video-based decision-making system and achieving satisfactory performance, I find the design of Stage 2 to be lacking in interpretability (If I am mistaken, please correct me). Stage 2 does not appear to sufficiently utilize the knowledge learned in Stage 1. The paper employs an adapter to cross-attend with the encoded features of historical images in open-sora, which can be viewed as extracting a latent representation from historical data, especially since the open-sora layers are also trained. Learning a latent representation can be accomplished in various ways, not limited to using an adapter. As a result, comparison with some representation learning baselines (e.g., R3M) is encouraged. Therefore, I suggest the authors explain what useful information is contained in these features and why cross-attention with these features is sufficient to distill the knowledge learned in Stage 1.

[1] Large-Scale Actionless Video Pre-Training via Discrete Diffusion for Efficient Policy Learning, 2024

**Questions:**

- The authors claim to reduce the open-sora model to 12 layers with 80M parameters. How is the 12-layer model initialized with the pretrained 700M weights? Do the authors simply select 12 layers and load the original weights?

**Limitations:**

This paper has discussed the limitations.

---

> ### Author Response · Authors · 2024-08-07
> **Response to Reviewer 5fDE [1/2]**
>
> Dear Reviewer,
>
> Thanks for your constructive comments. Below, we would like to address all the weaknesses and questions in details.
>
> **Q1:** `Ablate the choices of adaptation, such as an output projector.`
>
> **A1:** In our approach, we've adopted layer-wise adapters because current literature supports that in-model parameter adjustment methods are the most parameter-efficient ways [2, 3] to fine-tune models. While it is also feasible to transfer a pre-trained model using non-intrusive techniques—such as prefixing tokens to the input or appending a projector to the output— these methods tend to require a larger volume of fine-tuning data and parameters to match the performance of in-model adapters. Furthermore, these non-intrusive methods primarily alter the model's behavior by adjusting the input/output levels, rather than directly influencing the model's core competency in aggregating and inferring knowledge across long-range cross-tokens. The latter capability is considered a fundamental enhancement of the Transformer architecture[6] over traditional RNN and LSTM models. To see the benefit of our layerwise cross-attention adapter, we also include a new set of experiments in Table (2c), where we have removed the layer-wise adapter and instead appended a learnable token to the observation tokens (similar to the cls token in ViT). To further validate the effectiveness of our layer-wise adapter, in the second stage, we pooled the output tokens and then passed them through an MLP to project them into a 2x1152 tensor. This tensor was then fed into the action head. The results are shown in the table below. Our approach outperforms pooling MLP projector and concatenating tokens by 8.0% and 4.0% on RLBench-10 benchmark, respectively.
>
> | **Type**              | **RLBench-10** |
> |-----------------------|----------------|
> | pooling mlp projector | 45.1           |
> | w/o adapter           | 53.1           |
> | layer-wise adaptor    | 57.1           |
>
> Furthermore, since the Open-Sora model we've adopted wasn't pre-trained with explicit latent planning as described in [1], we don't presume that its hidden representations can be utilized in the same manner. We've observed that the diffusion planning in the latent space, as presented in [1], is computationally expensive and still fundamentally relies on the model's hidden representations. Given the Transformer architecture's demonstrated strong reasoning capabilities, we respectively don't agree that there's a compelling need to engage in explicit planning based on the hidden representations, as done in [1].
>
> **Q2:** `Comparison with some representation learning baselines (e.g., R3M) and explain what useful information is extracted in the representation and why cross-attention with these features is sufficient.`
>
> **A2:** Thank you for your constructive feedback. We've taken your suggestions on board and have now included comparative experiments with R3M in the following table. Before delving into the results, we'd like to distinguish between our approach and the latent representation learning, e.g., R3M. As previously mentioned in our last response, we've incorporated a cross-attention adapter to directly modulate the Transformer model's attention mechanism. This adjustment aims to enhance the effectiveness of the modified cross-attention by training the Open-Sora layers and aggregating more valuable information via the fine-tuned adapter, thereby reducing loss. In contrast to our method, R3M and similar approaches apply direct supervision to the latent representation using optimization techniques such as contrastive learning. They impose manual regularization on the learned representations, which may give the appearance of greater 'interpretability'. We employ the reproduction of R3M from GR-1 for comparison on the CALVIN benchmark, which uses R3M to encode observation images and leverages a GPT-style transformer to output actions. The comparison results between ours and R3M are shown as follows.
>
> | **CALVIN** | **1** | **2** | **3** | **4** | **5** | **Avg.Len.** |
> |------------|-------|-------|-------|-------|-------|--------------|
> | R3M(GR-1)  | 0.529 | 0.234 | 0.105 | 0.043 | 0.018 | 0.93         |
> | GR-1       | 85.4  | 71.2  | 59.6  | 49.7  | 40.1  | 3.06         |
> | Ours       | 95.9  | 81.6  | 73.5  | 61.2  | 55.1  | 3.673        |

---

> ### Author Response · Authors · 2024-08-07
> **Response to Reviewer 5fDE [2/2]**
>
> Regarding the pursuit of interpretability, we believe its value lies in the potential inductive bias it provides for purely data-driven methods, potentially enhancing performance. Ultimately, however, performance is the key metric that determines the usefulness of such 'interpretability.' In this regard, performance is the end goal, not interpretability itself. As a playful aside, the most powerful and enigmatically uninterpretable machine we know is, arguably, the human brain :).
>
> As for the discussion on the sufficiency of the cross-attention adapter we've adopted for distilling knowledge acquired in Stage 1, a few points must be clarified. First, using adapters to adjust how models behave is recognized in the literature [2, 3, 4, 5], as one of the most efficient ways. Notably, in robotics, studies like [2, 5] are using attention-based techniques to refine knowledge from transformer layers. We stand by our choice, confident that it is sufficient to achieve significant improvements over strong baselines, as evidenced by our experimental results. We've also tested out other non-cross-attention methods, such as directly using a projector, which appears to be less effective than ours. Our claims are based on our experiments and we're not claiming to have the ultimate method for squeezing every last bit of knowledge out of the Open-Sora model. And honestly, we're also skeptical that there will be a method, including those mentioned by the reviewer, that can claim their methods are already sufficient for their objectives.
>
> In conclusion, we believe that the interpretability of the representation and the absolute sufficiency of the cross-attention adapter should not be barriers to the approval of our article. Instead, these aspects should stimulate further research and discussion within
> the field.
>
> **Q3:** `How to select the Open-Sora layers.`
>
> **A3:** Sorry for the confusion. Open-Sora originally had 28 layers, but we only selected the first 12 layers, totaling 315M parameters. Our model has two deployment modes: one that only trains the layer-wise adapter and diffusion head, which has 80M parameters, and another that trains everything, totaling 395M parameters. This is a point that was not clearly explained in the text. We will improve it in the next revised version.
>
> References:
>
> [1] Large-Scale Actionless Video Pre-Training via Discrete Diffusion for Efficient Policy Learning, 2024.
>
> [2] Vision-Language Foundation Models as Effective Robot Imitators, 2023.
>
> [3] Flamingo: a Visual Language Model for Few-Shot Learning, 2022.
>
> [4] Unsupervised Cross-Modal Alignment for Multi-Person 3D Pose Estimation, 2020.
>
> [5] Octo: An Open-Source Generalist Robot Policy, 2023.
>
> [6] Attention is all you need, 2017.

---

> ### Comment · Reviewer_5fDE · 2024-08-08
> **Thanks for your hard work and response!**
>
> Firstly, I would like to remind you that you should post your rebuttal using the 'rebuttal' button instead of the 'official comment'. I have to reply to your response by commenting after my initial review to ensure the visibility of other reviewers.
>
> Overall, after reading your response and other reviews, I still have the following concerns:
> - While I appreciate your technical contribution and the proposed two-stage pipeline, which performs well in RLBench and other benchmarks (including the experimental results added during the rebuttal period), I have some questions about the performance in RLBench. **The success rates of the tasks 'screw bulb', 'place wine', 'sort shape', and 'push buttons' shown in the global PDF differ significantly from those in your original paper.** What caused these discrepancies? Did you revise your method?
> - This paper's main contribution lies in engineering, such as the incorporation of various modules like open-sora and diffusion policy. Both the modules used in this paper and the two-stage training design are not new. [1] and [2] use a similar two-stage training. For a paper to be accepted at NeurIPS, there should be some novel methods or insights to inspire future research. This is the main reason why I do not give a high rating.
>
> [1]  Large-Scale Actionless Video Pre-Training via Discrete Diffusion for Efficient Policy Learning, Arxiv 2024.
>
> [2]  Unleashing large-scale video generative pre-training for visual robot manipulation, ICLR 2024

---

> > ### Comment · Reviewer_yiak · 2024-08-08
> >
> > I agree with Reviewer 5fDE. Please make rebuttals responding to individual reviewers visible to all reviewers.

---

> ### Author Response · Authors · 2024-08-09
>
> Thank you for your response and further raised questions. The responses to the comments are as follows. We have made rebuttals visible to all reviewers.
>
> 1. The experiments in Table 1 of the attachment expand our method to multi-view cameras to make a fair comparison with RVT and Act3D. Note that, our method and RVT differ in many aspects, including the model architecture, inputs, and outputs, please refer to the **#A1** response to the Reviewer yiak. Although an apple-to-apple comparison is difficult, our method can be easily extended to handle multi-view sensory input, To achieve this, we only need to replace our overhead camera input with multiview cameras with the same resolution 128 x 128 (including front camera, left shoulder camera, right shoulder camera, and wrist camera ) as the input of STDiT in the 2nd stage. **We found that adding additional multi-view camera inputs significantly improves performance on tasks involving small objects often occluded by the robotic arm, such as buttons, blocks, and bulbs, or tasks requiring placing objects in specific locations, such as racks.** This could explain why our method shows noticeable improvements in these tasks, such as 'screw bulb', 'place wine', 'sort shape', and 'push buttons'.
>
> 2. The reviewer has noted that the two-stage training approach is a common practice for developing representations in action prediction tasks. However, we would like to clarify that we view the two-stage training as a training framework rather than a motivation to conclude either others or our method. In this regard, we respectfully disagree with the reviewer's oversimplification of our motivations and then equating it with the two-stage training pipeline.
>
> - We must point out that our motivation is fundamentally different from GR-1 [1]: As stated in the GR-1 paper, its goal is to verify that video datasets are beneficial to robot action learning, so their motivation guided their research work; whereas we believe that learning in System 2 is beneficial for modulate the precise execution of System 1, and combining the dual-process theory, we propose that slow thinking (denoising diffusion) in Stage-1, which is carefully design to enable the fast execution (one-pass inference) in Stage-2 of our method, thus designing the model. It is not difficult to find that our training goals and the model choices for each phase are closely linked to the motivation.
> - As for VPDD [2], its main contribution lies in the first stage, where a mask-and-replace diffusion strategy is proposed to predict actionless videos, while the second stage still predicts both actions and videos simultaneously, which can often slow down real-time inference during deployment. Our core motivation, however, is focused on slow thinking and fast execution, which perfectly aligns with the intuitive thinking of robotic cognition and behavior. Additionally, our method leverages the Open-Sora STDiT architecture and utilizes a layer-wise adapter to integrate the intrinsic dynamic knowledge from videos. This gives our model strong scalability and extensibility, allowing it to be effectively trained on large-scale robotic trajectory datasets like OXE or human demonstrations like Ego4D, and easily extended to multi-view inputs. VPDD has not been published in an official conference or journal, but its method has some valuable insights. We will discuss the differences between our approach, GR-1, and VPDD in the revised version.
>
> References:
>
> [1] Unleashing large-scale video generative pre-training for visual robot manipulation, ICLR 2024
>
> [2] Large-Scale Actionless Video Pre-Training via Discrete Diffusion for Efficient Policy Learning, Arxiv 2024.

---

> > ### Author Response · Authors · 2024-08-10
> >
> > Dear Reviewer,
> >
> > I hope this message finds you well. I wanted to kindly follow up on the response we provided to your latest comments. We appreciate the time and effort you have already dedicated to reviewing our paper, and your feedback has been invaluable in helping us refine our work.
> >
> > We are confident that our latest response has adequately addressed all your concerns. With the recent acknowledgment from Reviewer yiak and Yu5P, we believe that the contributions of our paper are now both technically sound and distinct from other works in the field. While we understand that you may still have reservations, we have faith that the overall contribution has met the standards required by NeurIPS.
> >
> > In light of this, if you agree, we believe our revised submission merits a higher score than the original version. We would be immensely grateful to see this recognition.
> >
> > Should there be any further questions or concerns, we are more than willing to address them. We recognize the significance of a thorough discussion and are keen to provide any additional clarification needed.
> >
> > Please let us know if there's anything more we can do to assist in this process. We are committed to ensuring that all aspects of our paper are fully understood and vetted.

---

> > > ### Comment · Reviewer_5fDE · 2024-08-11
> > >
> > > Thanks for your answers to address my concerns. I'd like to raise my score to borderline accept.

---

> > > > ### Author Response · Authors · 2024-08-11
> > > >
> > > > Thank you very much for reconsidering our paper and for raising your score. We appreciate your acknowledgment of our efforts to address your concerns.
> > > >
> > > > Your feedback has been invaluable in improving the quality of our submission, and we are grateful for the time and thoughtfulness you have invested in your review.

---

### Official Review · Reviewer_NqNP · 2024-07-10

**Soundness:** 3
**Presentation:** 3
**Contribution:** 3
**Rating:** 5
**Confidence:** 4

**Summary:**

This paper proposes a framework for video diffusion in robot manipulation (VidMan), which contains two stages. In the first stage, VidMan adopts large-scale Open-X datasets for pertaining an Open-Sora-like architecture for video prediction. In the second stage, an adapter is introduced to learn the inverse dynamics for policy learning. The whole framework can be used for robot manipulation tasks such as RLBench.

**Strengths:**

1. The whole framework adopts advanced techniques like video diffusion, open-sora, and adapter, which are popular for LLM/VLM-style research. Such a framework can motivate related research on introducing LLM/VLM architecture in robot manipulations.

2. The framework is motivated by dual process theory, which is reasonable.

**Weaknesses:**

1. The whole framework is very similar to small-scale diffusion planners like decision-diffuser, i.e., trajectory diffusion for planning and inverse dynamics for action prediction. Indeed, VidMan has unique contributions for scaling, but the basic concepts are similar to previous methods.

2. The existing framework requires high-quality expert trajectories in pertaining. If the non-optimal data is contained in the dataset, the video diffusion model may generate sub-optimal trajectories, and the resulting actions may also be non-expert. Can you discuss more about how to extend such a framework to sub-optimal datasets.

3. Some empirical results are not convincing. Please refer to the questions.

**Questions:**

1. Is the video tokenizer directly adopted from a pre-trained model? It may be better to fine-tune it with Open-X robot data.

2. In Table 1, I find that PerAct's results are lower than those of the original paper; can you explain this?

3. RVT is a multi-view benchmark that often requires multi-view images as input. Can VidMan generate multi-view images? Are the results in Table 1 learned by single-view images?

---

> ### Author Response · Authors · 2024-08-07
> **Response to Reviewer NqNP [1/2]**
>
> Dear Reviewer,
>
> Thanks for your constructive comments. Below, we would like to address all the weaknesses and questions in details.
>
> **Q1:** `Similar with diffusion-diffuser.`
>
> **A1:** For trajectory diffusion for planning and inverse dynamics for action prediction, this should be the most intuitive research path for methods that infer actions from videos. Decision-diffuser uses classifier-free guidance with low-temperature sampling to generate a sequence of future states. It then uses inverse dynamics to extract and execute the action that leads to the immediate future state. We emphasize that we have the following distinct contributions. a. we exploit "intrinsic" dynamics from the diffusion model, i.e., we do not explicitly infer the next state from the previous state; instead, it is done implicitly. Decision-diffuser is explicit, which is not suitable for real-time robotic manipulation tasks. b. We emphasize our design specifically for utilizing large amounts of robotic video data, such as the two-stage training method and the layer-wise adapter.
>
> **Q2:** `Discussion about how to extend our method to sub-optimal datasets.`
>
> **A2:** To verify the effectiveness of our method on sub-optimal trajectories, we replaced the first stage training dataset with Ego4D[1]. Ego4D is the world's largest egocentric (first person) video dataset, with 3,600 hrs (and counting) of densely narrated video. Ego4D has a smaller domain gap compared to internet data for robotic manipulation tasks and is easier to collect than high-quality expert trajectory datasets. With the above features, Ego4D can easily serve as a non-optimal dataset. The results are shown in the flowing table. When the domain gap of the pre-training dataset is close to that of the robotic manipulation task, the performance of our second-stage training also improves.
>
> | **Type**   | **RLBench-10** |
> |------------|----------------|
> | w/ webdata | 51.2           |
> | w/ Ego4D   | 55.5           |
> | w/ OXE     | 57.1           |
>
> **Q3:** `Whether fine-tune video tokenizer or not.`
>
> **A3:** In our implementation, we have chosen not to fine-tune the video tokenizer in order to minimize training costs. The question of whether to fine-tune the tokenizer of a pre-trained model has been a topic of interest in the field of parameter-efficient fine-tuning. These researches[3, 5, 7] have indicated that employing in-model adapters is often a more parameter-efficient approach and is less likely to compromise the original capabilities of the pre-trained model. While adjusting the tokenizer can be beneficial when dealing with extensive datasets, it may also lead to overfitting and could easily alter the function of the pre-trained model in cases that the finetuning data are too different from the pretraining data. Consequently, many methods that leverage pre-trained models, such as MAE[2] in GR-1[3], CLIP[4] in Octo[5], and DinoV2[6] in OpenVLA[7], opt not to modify the pre-trained tokenizer.
>
> **Q4:** `PerAct's results are lower than those of the original paper.`
>
> **A4:** We checked the results reported in our paper and compared them with those in the PerAct paper and found that they are consistent. Additionally, we found that the PerAct results reported in RVT are higher than those in the original PerAct paper, which may have caused some confusion. The RVT paper states that they reevaluated the PerAct model weights in their experimental environment, which resulted in higher performance. The following table shows the results from the original PerAct paper and the reproduced results by RVT.

---

> ### Author Response · Authors · 2024-08-07
> **Response to Reviewer NqNP [2/2]**
>
> | **Source**      | **opendrawer** | **slide block** | **sweep to dustpan** | **meat off grill** | **turn tap** | **put in drawer** | **close jar** | **drag stick** | **stack blocks** | **screw bulb** |
> |-----------------|----------------|-----------------|----------------------|--------------------|--------------|-------------------|---------------|----------------|------------------|----------------|
> | Official        | 80             | 72              | 56                   | 84                 | 80           | 68                | 60            | 68             | 36               | 24             |
> | Reported in RVT | 88.0 ± 5.7     | 74.0 ± 13.0     | 52.0 ± 0.0           | 70.4 ± 2.0         | 88.0 ± 4.4   | 51.2 ± 4.7        | 55.2 ± 4.7    | 89.6 ±         | 26.4 ± 3.2       | 17.6 ± 2       |
> | **Source**      | **put in safe** | **place wine** | **put in cupboard** | **sort shape** | **push buttons** | **insert peg** | **stack cups** | **place cups** | **Avg. Success** | **-** |
> | Official        | 44              | 12             | 16                  | 20             | 48               | 0              | 0              | 0              | 62.8             | -     |
> | Reported in RVT | 84.0 ± 3.6      | 44.8 ± 7.8     | 28.0 ± 4.           | 16.8 ± 4.7     | 92.8 ± 3.0       | 5.6 ± 4.       | 2.4 ± 2        | 2.4 ± 3.2      | 49.4             | -     |
>
> **Q5:** `Compared with RVT.`
>
> **A5:** We have identified several challenges in making a direct comparison with our approach. Firstly, RVT leverages four RGB-D cameras, significantly enhancing environmental perception capabilities. Our method, however, adopts a more cost-effective and widely applicable setup, typically utilizing just an overhead camera, sometimes complemented by a wrist camera. Secondly, RVT focuses solely on predicting target positions and rotation angles, delegating motion planning to a separate system, whereas our method offers end-to-end control. Both approaches have their merits, with RVT highlighting the value of advanced sensing and our method emphasizing embodied control. As shown in the following table, without the need for a time-consuming rendering process and explicit motion planning, our method is much more efficient during inference compared to RVT. Despite the differences, we see no inherent conflict between our method and RVT. In fact, our approach is adaptable to various sensor inputs, which has the potential to enhance performance. To illustrate, we have standardized the first issue by incorporating multiview inputs into our method. Specifically, we have replaced the overhead camera input with a set of multiview cameras—each with a resolution of 128x128, including front, left shoulder, right shoulder, and wrist cameras—as used by STDiT in the second stage. The results are shown as follows.
>
> | **Models**        | **opendrawer** | **slide block** | **sweep to dustpan** | **meat off grill** | **turn tap** | **put in drawer** | **close jar** | **drag stick** | **stack blocks** | **screw bulb** |
> |-------------------|----------------|-----------------|----------------------|--------------------|--------------|-------------------|---------------|----------------|------------------|----------------|
> | RVT               | 71.2 ± 6.9     | 81.6 ± 5.4      | 72.0 ± 0.0           | 88.0 ± 2.5         | 93.6 ± 4.1   | 88.0 ± 5.7        | 52 ± 2.5      | 99.2 ± 1.6     | 28.8 ± 3.9       | 48.0 ± 5.7     |
> | VidMan(multiview) | 94.4 ± 3.6     | 97.6 ± 4.4      | 92.8 ± 1.8           | 90.4 ± 2.2         | 96.8 ± 3.3   | 83.2 ± 1.8        | 88 ± 2.8      | 84.8 ± 3.5     | 48 ± 0.0         | 66.4 ± 2.1     |
> | **Models**        | **put in safe** | **place wine** | **put in cupboard** | **sort shape** | **push buttons** | **insert peg** | **stack cups** | **place cups** | **Avg score** | **Inf. Speed** |
> | RVT               | 91.2 ± 3.0      | 91.0 ± 5.2     | 49.6 ± 3.2          | 36.0 ± 2.5     | 100.0 ± 0.0      | 11.2 ± 3.0     | 26.4 ± 8.2     | 4.0 ± 2.5      | 62.9          | 11.6           |
> | VidMan(multiview) | 67.2 ± 3.3      | 79.6 ± 0.0     | 32.8 ± 3.3          | 48.0 ± 0.0     | 89.6 ± 1.6       | 21.6 ± 3.6     | 18.2 ± 1.8     | 13.2 ± 1.8     | 67.4   | 18.3           |
>
> References:
>
> [1] Ego4d: Around the world in 3,000 hours of egocentric video.
>
> [2] Masked Autoencoders Are Scalable Vision Learners.
>
> [3] Unleashing Large-Scale Video Generative Pre-training for Visual Robot Manipulation.
>
> [4] Learning Transferable Visual Models From Natural Language Supervision.
>
> [5] Octo: An Open-Source Generalist Robot Policy.
>
> [6] DINOv2: Learning Robust Visual Features without Supervision.
>
> [7] OpenVLA: An Open-Source Vision-Language-Action Model.

---

> > ### Author Response · Authors · 2024-08-10
> >
> > Dear Reviewer,
> >
> > I hope you are doing well. I wanted to kindly follow up on the response we provided to your insightful comments. Your feedback has been instrumental in shaping our work, and we deeply value the time and expertise you have dedicated to reviewing our paper.
> >
> > If there are any further questions or if you would like to discuss any of the points raised in your initial review, we would be eager to continue the conversation. We are committed to addressing any concerns and ensuring that our work meets the highest standards of quality and clarity.
> >
> > Thank you once again for your thoughtful review. We look forward to any additional feedback you may have.

---

> > > ### Comment · Reviewer_NqNP · 2024-08-10
> > > **Questions**
> > >
> > > Thanks for the response. I have additional questions concerning the new experiments.
> > >
> > > 1. "we do not explicitly infer the next state from the previous state; instead, it is done implicitly", can you explain more?
> > >
> > > 2. Indeed, Ego4D can be considered as sub-optimal data but with the dynamics shift, which makes it is challenging to directly use for robotics tasks. About the Ego4D experiment, which subset of the Ego4D dataset is used for training? what about the amount of data compared to other settings?
> > >
> > > 3. How to replace the overhead camera input with a set of multiview cameras? Does it change the architecture of your methods? How to ensure multi-view consistency? If each state includes front, left shoulder, right shoulder, and wrist cameras, the videos/embeddings (e.g., $V_s$) used for training have much larger dimensions than the original one, how to handle this challenge?

---

> ### Author Response · Authors · 2024-08-11
>
> Thank you for your additional questions and for the opportunity to further clarify our work.
>
> 1. To clarify, at Stage 2 of our method, we do not directly predict future frames based on past frames when predicting future actions. Specifically, through the first stage of training, our model acquires the ability to predict future frames. In the second stage,  we adapt the model to learn to directly predict the actions given the historical observation and task instruction, without the need for iteratively decoding frames explicitly. This allows the model to effectively acquire fine-grained real-world dynamics via the multi-step video denoised diffusion at the first stage, and efficiently leverage such learned dynamics to predict actions by adapting the model to function as a one-pass inverse dynamics model. In other words, the image observations are encoded into latent representations which may include information about future frames due to the pretraining objective. However, explicit future predictions, either latent or pixel, are not explicitly generated. An action is computed based on these inputs.
>
> 2. a) We did not directly use Ego4D for robotic action prediction. Instead, we employed them in the first stage of video prediction pre-training. This allowed our model to acquire the intrinsic dynamic knowledge embedded in the videos. Similar to ours, methods such as [R3M, GR-1] have also utilized Ego4D to acquire its intrinsic dynamics. Therefore, using Ego4D for learning prior knowledge of robot control in common and rationale in the field. b) We used the forecasting Hands and Object (FHO) annotations from the Ego4D dataset. FHO captures how the camera-wearer changes the state of an object by using or manipulating it. For a single video, it might include two narration processes: narration_pass_1 and narration_pass_2. We segmented the video data based on these two narration passes and filtered out clips longer than 300 seconds and shorter than 30 seconds. In the end, we obtained approximately 283k and 340k clips, totaling around 623k clips. OXE dataset contains about 800k robot trajectories. Regarding the web data, we only loaded the Open-Sora weights and did not perform any training on the web data itself. Due to our iterative training technique, with a batch size of 384 and 100k training iterations in the first stage, we believe the model has been adequately trained on both the OXE and Ego4D datasets. We will expose all of the data processing details and training details in our revision in case of interest.
>
> 3. Since our method is based on the Open-Sora STDiT architecture, instead of changing the architecture, we treat multiview images as tokens and input them into the transformer. Following [RVT], tokens belonging to the same image are processed via the first four transformer layers for each image. Finally, the processed image tokens are jointly processed using the last eight transformer layers. This attention masking approach can automatically maintain multi-view consistency which is also employed by RVT. In our original paper, our model takes in a single-view image of size 256×256 with a patch size of 16, resulting in a total of 256 tokens for the image. To extend the model to handle four images, we resize each image to have the size 128×128, also resulting in 256 tokens in total for images. Thus, this change does not slow down the training or inference time.
>
> We hope this provides clarity on the concerns you raised. All of the mentioned details will be provided in our revised manuscript to ensure that our work is transparent and robust, and we appreciate your continued engagement and feedback.
>
> References:
>
> [R3M] R3M: A Universal Visual Representation for Robot Manipulation.
>
> [GR-1] Unleashing Large-Scale Video Generative Pre-training for Visual Robot Manipulation.
>
> [RVT] Robotic view transformer for 3d object manipulation.

---

> > ### Comment · Reviewer_NqNP · 2024-08-13
> >
> > Thanks for the response,  the technique "resize each image to have the size 128×128, also resulting in 256 tokens in total for images" is interesting in predicting multi-view images. I raise my score accordingly.

---

> > > ### Author Response · Authors · 2024-08-13
> > >
> > > Dear Reviewer,
> > >
> > > Thank you very much for your positive feedback and for raising your score. We are glad to hear that you found our approach to resizing images for multi-view prediction interesting.
> > >
> > > Your insights and suggestions have been instrumental in refining our work, and we truly appreciate the time and effort you have invested in this review process.
> > >
> > > Thank you once again for your support.

---

> ### Author Response · Authors · 2024-08-13
> **Follow-up on Response for Review Comments**
>
> Dear Reviewer,
>
> I hope this message finds you well. I wanted to kindly follow up on the detailed response we provided to your recent comments. We greatly appreciate the time and effort you have already invested in reviewing our paper, and your insights have been instrumental in refining our work.
>
> We believe that our latest response has addressed your concerns comprehensively. With the recent acknowledgment from Reviewer yiak, Yu5P and 5fDE, we are confident that our paper’s contributions are both technically robust and distinct within the field. While we understand that you may still have reservations, we hope you agree that our revised submission meets the high standards expected by NeurIPS.
>
> In light of these revisions and the feedback from other reviewers, we believe our submission merits a higher score than the original version, and we would be grateful if this could be reflected in your final assessment.
>
> If there are any additional questions or concerns, we are more than willing to address them promptly. We are committed to ensuring that every aspect of our work is fully understood and vetted.
>
> Please let us know if there’s anything more we can do to assist in this process. Your continued feedback is invaluable, and we are eager to provide any further clarification you may need.
>
> Thank you once again for your support and consideration.

---

### Official Review · Reviewer_11a6 · 2024-07-10

**Soundness:** 3
**Presentation:** 3
**Contribution:** 3
**Rating:** 7
**Confidence:** 3

**Summary:**

This paper presents a method called Video Diffusion for Robot Manipulation (VidMan) that uses a two-stage training mechanism. In the first stage, the VidMan model is trained to perform future state prediction (i.e. dynamics learning) on a large diverse dataset (Open X-Embodiment). In the second stage, the model is finetuned using a cross-attention adaptation layer to instead learn an inverse dynamics model. The method is evaluated on RLBench tasks, outperforming baselines, and on offline OXE data. The authors further provide ablation experiments to illustrate the effect of each of the components of the method.

**Strengths:**

- The work addresses an important and timely problem, which is how to leverage large-scale data to learn dynamics, and then propagate that information to downstream policy learning.
- The empirical results of the method seem strong, in particular on the RLBench tasks.
- Ablations demonstrate that training the second stage on an action-only objective significantly improves performance compared to co-training.

**Weaknesses:**

- I think the work could better highlight the validation of the central hypothesis of the work that the two-stage training process is actually more effective than a single stage training process (just learning the inverse dynamics model). Although the authors provide a baseline that compares the performance when using a co-training objective in stage 2 of training, if I understand correctly, the ablation that does not perform stage 1 of training at all and directly learns an inverse model is the one found in Table 2(b) “w/o domain train”. I think as this is one of the main points that the work is trying to argue for, this should be highlighted to a greater degree. Directly comparing the entire VidMan system to models like Octo or PerAct can’t really validate this hypothesis because those methods are so different. Additional experiments to test this on the offline evaluation setting would also help.
- The baselines selected for the RLBench task seem somewhat arbitrary – For instance, robotic view transformer (RVT) by Goyal et al. in 2023. significantly improves upon PerAct in the RLBench task suites, and RVTv2 improves upon that further.
- The manuscript has several grammatical errors that do not interfere with the reader’s understanding of the work (e.g. L292 “Whether pretrained with general video data on the internet help improve the performance”), but I suggest the authors carefully revise the manuscript.
- The connection to the two-process model for cognitive systems seems rather tangential, and I would recommend the authors to omit it in favor of more grounded discussion about the concrete implementations of the training stages.

**Questions:**

- Can the authors clarify what the term “intrinsic dynamics” means?
- In L284, “This improvement is attributed to our approach’s ability to utilize the training data more efficiently by implicitly leveraging the inductive bias of inverse dynamics”, should this be “forward dynamics” instead of “inverse dynamics”?

**Limitations:**

The authors give a brief description of the limitations, which is rather terse. I would appreciate it if the authors could address additional limitations of their work from various aspects.

---

> ### Author Response · Authors · 2024-08-07
> **Response to Reviewer 11a6 [1/2]**
>
> Dear Reviewer,
>
> Thanks for your constructive comments. Below, we would like to address all the weaknesses and questions in details.
>
> **Q1:** `In-depth analyses for the effectiveness of the adopted two-stage strategy.`
>
> **A1:** We greatly appreciate the thoughtful feedback you've provided. In our initial manuscript, we carried out an array of experiments designed to underscore the efficacy of our two-stage strategy. Upon review, we've come to understand that the organization of our experimental findings did not fully convey the comprehensive benefits of our approach. Furthermore, it has come to our attention that there are supplementary experiments that could offer more explicit validation of our methods' effectiveness. In the following content, we are committed to incorporating the existing and newly added findings to highlight that the strengths of our adopted two-stage strategy.
>
> a. Is two-stage training better than single-stage in the final result?
> Based on the reviewer's suggestion, we also conducted experiments in an offline evaluation setting. The experimental setups for 1-stage and 2-stage are as follows: for 1-stage, action prediction is trained on OXE for 300K steps. For the 2-stage, video prediction is trained on OXE for 100K steps in the first stage, and action prediction is trained on OXE for 200K steps in the second stage. Finally, both are evaluated offline on the Bridge dataset. The results are shown as follows. Additional experiments showed similar performance to the original experiments conducted on RLBench-10 (the leftmost column.), proving that 2-stage training is more effective than 1-stage training.
>
> | **Type** | **RLBench-10** | **MSE** | **xyz** | **angle** |
> |----------|----------------|---------|---------|-----------|
> | 1-stage  | 51.2           | 1.33    | 48.5    | 52.4      |
> | 2-stage  | 57.1           | 0.95    | 50.1    | 56.8      |
>
> b. Can pretrained with non-robotic data, e.g. Ego4D, at the first stage still possess a similar advantage against the single-stage alternative?
> We replaced the training dataset in the first stage with EGO4D (a first-person video dataset of human-object interactions), which also showed performance improvement compared to 1-stage training. The results are shown in the table below. This demonstrates that in 2-stage training, what is needed is to learn the intrinsic representation of dynamics from the video prediction training in the first stage. The training dataset in the first stage is dynamic but non-robotic, and it also benefits the performance of action prediction.
>
> | **Type** | **RLBench-10** |
> |----------|----------------|
> | 1-stage  | 51.2           |
> | w/ Ego4D | 55.5           |
> | w/ OXE   | 57.1           |
>
> c. How does our method compare to other two-stage methods, such as GR-1?
> GR-1 also employs a two-stage training approach and achieves superior performance on CALVIN. In the first stage, it uses Ego4D for video generation pre-training. In the second stage, it generates actions and the next frame using the next token generation method. For a fair comparison, we replaced our first-stage dataset with Ego4D and compared our model with GR-1 on the CALVIN benchmark. Experiments show that our method outperforms some two-stage methods, which are shown as follows.
>
> | **CALVIN**   | **1** | **2** | **3** | **4** | **5** | **Avg.Len.** |
> |--------------|-------|-------|-------|-------|-------|--------------|
> | GR-1         | 85.4  | 71.2  | 59.6  | 49.7  | 40.1  | 3.06         |
> | Ours (Ego4D) | 88.7  | 77.5  | 63.8  | 54.1  | 45.3  | 3.29         |
>
> **Q2:** `Comparison with RVT. `
>
> **A2:** Thanks for your suggestion. Directly comparing our method with RVT is not entirely fair. Our proposed method has the following key differences from RVT and other state-of-the-art approaches: 1. RVT utilizes four RGB-D cameras, including a front camera, left shoulder camera, right shoulder camera, and wrist camera. In contrast, our method only uses an overhead camera and a wrist camera, and it does not incorporate depth information. 2. RVT predicts the position and rotation angles of keyframes, then uses a sampled motion planner to move the robotic arm to the corresponding position and rotate to the corresponding angle. Our method, on the other hand, outputs a 6-dimensional vector end-to-end, which is directly executed by the end-effector controller. An absolutely fair comparison may not be entirely feasible, but we can adopt some relatively fair comparisons. Specifically, we have standardized the first difference mentioned above by also using multiview input in our proposed method. Fortunately, due to the flexibility of our model, we only need to replace our overhead camera input with multiview cameras with the same resolution 128 x 128 (including front camera, left shoulder camera, right shoulder camera and wrist camera ) as the input of STDiT in the 2nd stage. The results are shown as follows.

---

> ### Author Response · Authors · 2024-08-07
> **Response to Reviewer 11a6 [2/2]**
>
> | **Models**        | **opendrawer** | **slide block** | **sweep to dustpan** | **meat off grill** | **turn tap** | **put in drawer** | **close jar** | **drag stick** | **stack blocks** | **screw bulb** |
> |-------------------|----------------|-----------------|----------------------|--------------------|--------------|-------------------|---------------|----------------|------------------|----------------|
> | RVT               | 71.2 ± 6.9     | 81.6 ± 5.4      | 72.0 ± 0.0           | 88.0 ± 2.5         | 93.6 ± 4.1   | 88.0 ± 5.7        | 52 ± 2.5      | 99.2 ± 1.6     | 28.8 ± 3.9       | 48.0 ± 5.7     |
> | VidMan(multiview) | 94.4 ± 3.6     | 97.6 ± 4.4      | 92.8 ± 1.8           | 90.4 ± 2.2         | 96.8 ± 3.3   | 83.2 ± 1.8        | 88 ± 2.8      | 84.8 ± 3.5     | 48 ± 0.0         | 66.4 ± 2.1     |
> | **Models**        | **put in safe** | **place wine** | **put in cupboard** | **sort shape** | **push buttons** | **insert peg** | **stack cups** | **place cups** | **Avg score** | **Inf. Speed** |
> | RVT               | 91.2 ± 3.0      | 91.0 ± 5.2     | 49.6 ± 3.2          | 36.0 ± 2.5     | 100.0 ± 0.0      | 11.2 ± 3.0     | 26.4 ± 8.2     | 4.0 ± 2.5      | 62.9          | 11.6           |
> | VidMan(multiview) | 67.2 ± 3.3      | 79.6 ± 0.0     | 32.8 ± 3.3          | 48.0 ± 0.0     | 89.6 ± 1.6       | 21.6 ± 3.6     | 18.2 ± 1.8     | 13.2 ± 1.8     | 67.4   | 18.3           |
>
> **Q3:** `The meaning of "intrinsic dynamics" and "inverse dynamics".`
>
> **A3:** Intrinsic dynamics encompass the world dynamics that a model implicitly learns from video data, which includes both forward and inverse dynamics. The forward dynamics, our primary focus in the first stage, describe how observations evolve from one moment to the next, essentially a temporal sequence prediction task. In our specific application, this refers to the diffusion-based process of video generation. Inverse dynamics, on the other hand, pertain to the inference of an action given consecutive observations (e.g. images) before and after the action is executed. In our context, this involves action prediction process with the aid of the video generator at the 2nd stage. Since we are validating that the video generation model can effectively infer actions, it indicates that the video model can provide a good inductive bias for inverse dynamics.
>
> To be more concise, our approach to efficiently infer action does not involve explicitly predicting frames. Instead, we predict actions based on the hidden representations within the video diffusion generator. To this end, our method relies on intrinsic inverse dynamics, utilizing the model's latent knowledge to forecast actions, rather than relying on explicit inverse dynamics that predict actions from generated images.
>
> **Q4:** `Discuss more about the limitations.`
>
> **A4:** Thank you for your suggestions.
> We will expand the limitations as follows:
> 1) Our current model operates solely on 2D vision and does not possess 3D perception capabilities. This limitation affects its performance on tasks that demand accurate 3D spatial understanding. Our future work will prioritize the integration of 3D perception into the model.
> 2) Our model's capacity to comprehend complex human instructions is limited. It currently lacks the sophistication of a T5 language encoder. We are considering state-of-the-art Large Language Models (LLMs), such as LLama-3.1, as a promising avenue for future exploration to enhance these capabilities.
> 3) The model's perception is not fine-grained. It processes the entire image as a single input for action prediction. We believe that incorporating fine-grained auxiliary inputs, such as object-bounding boxes or masks, could significantly enhance the model's performance. We are exploring ways to integrate such detailed inputs to refine the model's perceptual abilities.

---

> > ### Author Response · Authors · 2024-08-10
> >
> > Dear Reviewer,
> >
> > I hope this message finds you well. I wanted to kindly follow up on the response we provided to your valuable comments and questions. Your feedback has been crucial in refining our work, and we greatly appreciate the time and effort you have invested in reviewing our paper.
> >
> > If there are any additional thoughts or questions regarding our responses, we would be more than happy to address them. We understand the importance of ensuring that all points are thoroughly discussed, especially concerning the central hypothesis and other key aspects you highlighted.
> >
> > Thank you once again for your insightful review, and we look forward to any further feedback you may have.

---

> > > ### Comment · Reviewer_11a6 · 2024-08-11
> > > **Response to authors**
> > >
> > > Dear Authors,
> > >
> > > Thank you for your great efforts in providing clarifications and additional experiments during the rebuttal period.
> > >
> > > The supplemental experiments provided in the rebuttal to evaluate the effect of the two-staged training pipeline are very helpful and successfully address my concerns on that point, and the added comparison to RVT also helps to make the paper significantly more complete.
> > >
> > > I wish to discuss more about the concept of "intrinsic inverse dynamics". To preface, I understand the concepts of forward and inverse dynamics models. Based on the description you provided, my understanding is that the "intrinsic inverse dynamics" takes as input text tokens and current image observations (not future observations). The image observations are encoded into latent representations which may include information about future frames due to the pretraining objective. However, explicit future predictions, either latent or pixel, are *not* explicitly generated. An action is computed based on these inputs.
> > >
> > > If this is the case, I feel that it's misleading to describe this process as "intrinsic inverse dynamics". There doesn't seem to be evidence that the model is necessarily incorporating information from potential future states. If there were such evidence, I think it would be more appropriate to call this "implicit inverse dynamics".
> > >
> > > In addition, I hope that we can clarify (or remove) the distinction between "intrinsic" and "implicit". These terms don't have the same meaning, but seem to be used interchangeably in this work. Based on your response that "Intrinsic dynamics encompass the world dynamics that a model implicitly learns from video data, which includes both forward and inverse dynamics," it seems like intrinsic dynamics are the same as implicit dynamics. If this is true, I recommend choosing one term and applying it consistently. I suggest "implicit dynamics", because the action prediction model is implicitly using knowledge about the dynamics, whereas it's less clear to me what "intrinsic dynamics" would mean without context.
> > >
> > > It's also not entirely clear what distinguishes intrinsic/implicit dynamics from just world dynamics, although in this case I can see that it is likely that dynamics is learned using implicit states rather than from raw pixels.
> > >
> > > At a high level, I agree with reviewer Yu5P on the points they made in their latest comment, and hope that their suggestions can be implemented.
> > >
> > > With those changes and after hopefully clarifying the discussion about the intrinsic/implicit dynamics as above, I am willing to increase my score. I appreciate the efforts the authors have made to provide clarifications and additional experiments in the rebuttal period.

---

> > > > ### Author Response · Authors · 2024-08-11
> > > >
> > > > Dear Reviewer,
> > > >
> > > > Thank you for your thoughtful comments and for engaging in a detailed discussion about the concepts of "intrinsic" and "implicit" dynamics. We greatly appreciate your feedback, which helps us refine our work further.
> > > >
> > > > You are correct in understanding that our model takes as input the text tokens and current image observations, with these observations encoded into latent representations that may include information about future frames due to the pretraining objective. However, as you pointed out, explicit future predictions are not directly generated, and the action is computed based on the inputs and the learned representations.
> > > >
> > > > We acknowledge that the term "intrinsic inverse dynamics" may be misleading in this context, as it could suggest that the model is explicitly incorporating information from potential future states. Instead, our model implicitly leverages the learned representations from the pretraining stage, which might encode dynamics information without generating explicit future states. Therefore, we agree that "implicit inverse dynamics" is a more accurate term to describe this process.
> > > >
> > > > We also appreciate your suggestion to standardize the terminology throughout the paper. We will revise the manuscript to consistently use the term "implicit dynamics" rather than "intrinsic dynamics," as this better captures the essence of the model's behavior. The term "implicit" emphasizes that the model is utilizing learned dynamics in a non-explicit manner, which aligns with the nature of our approach.
> > > >
> > > > We have taken reviewer Yu5P's comments into consideration and are in the process of implementing the suggested changes. We believe that these revisions, along with the clarification and standardization of our terminology, will enhance the clarity and impact of our work.
> > > >
> > > > Thank you again for your constructive feedback and for your willingness to increase your score following these changes. We are committed to making the necessary adjustments and appreciate your support in refining our paper.

---

> ### Author Response · Authors · 2024-08-13
>
> Dear Reviewer,
>
> I hope this message finds you well. I wanted to follow up regarding the response I sent earlier addressing your insightful comments on our paper, particularly the discussion around the "intrinsic" and "implicit" dynamics.
>
> We have carefully considered your feedback and have started implementing the suggested changes, including the revision of terminology to ensure clarity and consistency throughout the manuscript. We believe these revisions will significantly improve the quality and precision of our work.
>
> If you have any further comments or require additional clarification, please don't hesitate to let us know. We are committed to making the necessary adjustments and greatly appreciate your time and expertise.
>
> Thank you again for your valuable feedback, and we look forward to your thoughts on the revised manuscript.

---

> > ### Comment · Reviewer_11a6 · 2024-08-13
> > **Response to Authors**
> >
> > Dear authors,
> >
> > Thank you for the additional clarification and consideration of my suggestions about the "intrinsic" vs. "implicit" terminology. I might (lightly) recommend also considering to modify the title accordingly, although directly substituting "implicit" for "intrinsic" may sound a bit awkward -- perhaps a minor rework would help. Whether or not you choose to update the title, I've increased my score based on the many rebuttal updates to both my and other reviewers' concerns that have helped to make this submission much more solid.

---

> > > ### Author Response · Authors · 2024-08-13
> > >
> > > Dear Reviewer,
> > >
> > > Thank you for your thoughtful feedback and for considering our clarifications on the "intrinsic" vs.  "implicit" terminology.  We appreciate your suggestion regarding the title and will carefully consider a possible rework to ensure it accurately reflects the content while maintaining clarity.
> > >
> > > We are grateful for your increased score and the constructive comments that have significantly strengthened our submission.  Your insights have been invaluable in refining our work.
> > >
> > > Thank you once again for your time and support.

---

### Official Review · Reviewer_yiak · 2024-07-12

**Soundness:** 4
**Presentation:** 3
**Contribution:** 4
**Rating:** 9
**Confidence:** 4

**Summary:**

This paper introduces VidMan, a two-stage framework for robot manipulation using a pre-trained video diffusion model. The first stage involves training on an actionless video dataset to predict future visual trajectories using denoising diffusion. The second stage adapts this model into an inverse dynamics model, predicting actions from the learned dynamics using a smaller and faster diffusion-based action head to learn an action distribution implicitly conditioned on state transitions through the video prediction model. VidMan outperforms multiple strong baselines in a variety of environments, showing impressive generalization capabilities and good overall performance.

**Strengths:**

**Novel Approach:** The two-stage training approach is a deep insight that may have a large impact on the robot learning community. Separating the modeling of environment dynamics from action prediction may lead to further advancements in behavior cloning.

**State-of-the-Art Performance:** VidMan achieves substantial improvements over strong baselines, particularly in scenarios with limited data.

**Thorough Evaluation:** Comprehensive experiments on multiple benchmarks (RLBench, OXE) validate the performance of the method, and the approach is also supported by extensive ablation studies.

**Weaknesses:**

**Lack of Comparison to SOTA Baselines:** The paper lacks comparisons to state-of-the-art baselines like Robotic View Transformer (RVT), a direct follow-up to PerAct that presented competitive results on the same set of RLBench tasks chosen for this work. I recommend that this work be cited and included in the baselines. The work can be found here: https://arxiv.org/abs/2306.14896

**Qualitative Results:** While the quantitative results are thorough, the paper could benefit from more qualitative results (e.g. videos or a website) to illustrate the model's performance and capabilities in dynamic environments. The videos provided in the supplementary were of high quality, but it is difficult to tell whether the few provided videos are representative of the policy's performance. Additionally, the provided video prediction results in Figure 7 within the appendix primarily show static scenes. I would prefer results with more complex dynamics to be convinced that the video prediction model is performant.

**Scalability Concerns:** The performance improvement over baselines seems to diminish as the number of demonstrations increases. This raises the concern that the method may not continue to show increased performance at larger scales. Additionally, the authors show only marginal improvement from pre-training on internet-scale video data, suggesting that the method may not sufficiently utilize non-robot data.

**Lack of Real-World Experiments** The authors provide no experiments with their policy running on a physical robot. Additionally, it is not shown whether the policy can run in real-time. One suggestion could be to evaluate policies trained on the OXE dataset in the SIMPLER environments (https://arxiv.org/abs/2405.05941). I understand that this is not a reasonable request for the purposes of your rebuttal, but it may be a good choice for the camera-ready version of this work.

**Questions:**

**Inference speed:** How quickly does the model run in online scenarios, and what are the computational resource requirements during deployment?

**Missing tasks:** Why were specific tasks like "insert peg," "stack cups," and "place cups" omitted from the RLBench evaluation, when they are included in PerAct and RVT?

**Missing baseline:** Why were RVT results not included, given that they have results for the same set of tasks and outperform PerAct? I will consider updating my rating if this baseline is included.

**Intuition for Web Data Ablation:** Given that video pre-training with web-scale data only marginally improves performance (+0.8% in success rate), is this due to the domain gap between web data and simulation data, because the in-domain data closely aligns with the evaluation data, or a different reason?

**Accuracy metric:** Can you precisely explain what the accuracy metric is measuring? Are the actions discretized and classified, or is there some error threshold that warrants a successful prediction? This should be explained in Section 5.1.2.

**Limitations:**

The paper addresses some limitations, particularly in terms of scalability to 3D perception tasks, but further discussion is needed regarding the model's real-world deployment and computational efficiency

---

> ### Author Response · Authors · 2024-08-07
> **Response to Reviewer yiak [1/3]**
>
> Dear Reviewer,
>
> Thanks for your constructive comments. Below, we would like to address all the weaknesses and questions in details.
>
> **Q1:** ` Compare with more related SOTA methods.`
>
> **A1:** Thank you for your insightful suggestions. In response to the reviewers' feedback, we have expanded our comparison with more baseline methods and evaluated in SIMPLER environment in the revised manuscript, demonstrating that our method continues to excel under fair conditions. A particularly pertinent comparison is with GR-1, which employs next-token prediction autoregression pretraining on Ego4D data, achieving state-of-the-art results on the CALVIN benchmark. CALVIN is a challenging benchmark focusing on learning language-conditioned policy for long-horizon robot manipulation. We perform experiments on the ABC->D split of data. The comparison results on CALVIN are shown as follows. The average length in the last column, computed by averaging the number of completed tasks in a row of 5 in all the evaluated sequences, shows the long-horizon capability in a comprehensive way. Since our method can utilize the diffusion model's perception of the intrinsic dynamics of videos and is pre-trained on a large amount of expert-annotated trajectory, our model significantly improves the average length of successful tasks by 0.61.
>
> | **CALVIN** | **1** | **2** | **3** | **4** | **5** | **Avg. Len.** |
> |------------|-------|-------|-------|-------|-------|--------------|
> | GR-1       | 85.4  | 71.2  | 59.6  | 49.7  | 40.1  | 3.06         |
> | Ours       | 95.9  | 81.6  | 73.5  | 61.2  | 55.1  | 3.67        |
>
> And, we also conducted validation in the SIMPLER environment. SIMPLER is a simulation environment that includes two common setups: Google Robot and WidowX. We evaluated our model on 25 episodes per task, and the results are shown as follows.
> | **Google Robot** | **pick_coke_can** | **pick_object** | **move_near** | **open_drawer** | **close_drawer** | **place_in_closed_drawer** | **Avg. Score** |
> |--------------------------|-------------------|-----------------|---------------|-----------------|------------------|----------------------------|----------------|
> | RT-X-1                    | 0                 | 0               | 0             | 0               | 0.12             | 0                          | 0.02           |
> | Octo-small               | 0.24              | 0.04            | 0.04          | 0.04            | 0.32             | 0                          | 0.11       |
> | Octo-base                | 0.04              | 0.08            | 0             | 0               | 0.44             | 0                          | 0.09       |
> | ours                     | 0.32              | 0.12            | 0.04          | 0.12            | 0.4              | 0.08                       | 0.18           |
>
> | **WidowX** | **spoon_on_towel** | **carrot_on_plate** | **stack_cube** | **put_eggplant_in_basket** | **Avg. Score** |
> |--------------------|--------------------|---------------------|----------------|----------------------------|----------------|
> | RT-X-1               | 0                  | 0                   | 0              | 0.04                       | 0.01           |
> | Octo-small         | 0.48               | 0.08                | 0.04           | 0.6                        | 0.3            |
> | Octo-base          | 0.12               | 0.04                | 0              | 0.32                       | 0.12           |
> | ours               | 0.56               | 0.2                 | 0              | 0.6                        | 0.34           |
>
> When it comes to RVT, we've identified several challenges in making a direct comparison with our approach: Firstly, RVT leverages four RGB-D cameras, significantly enhancing environmental perception capabilities. Our method, however, adopts a more cost-effective and widely applicable setup, typically utilizing just an overhead camera and a wrist camera. Secondly, RVT focuses solely on predicting target positions and rotation angles, delegating motion planning to a separate system, whereas our method offers end-to-end control. Both approaches have their merits, with RVT highlighting the value of advanced sensing and our method emphasizing embodied control. As shown in the following table, without the need for a time-consuming rendering process and explicit motion planning, our method is much more efficient during inference than RVT. Despite the differences, we see no inherent conflict between our method and RVT. In fact, our approach is adaptable to various sensor inputs, which has the potential to enhance performance. To illustrate, we've standardized the first issue by incorporating multiview inputs into our method. Specifically, we've replaced the overhead camera input with a set of multiview cameras--each with a resolution of 128x128, including front, left shoulder, right shoulder, and wrist cameras--as used by STDiT in the second stage.

---

> ### Author Response · Authors · 2024-08-07
> **Response to Reviewer yiak [2/3]**
>
> | **Models**        | **opendrawer** | **slide block** | **sweep to dustpan** | **meat off grill** | **turn tap** | **put in drawer** | **close jar** | **drag stick** | **stack blocks** | **screw bulb** |
> |-------------------|----------------|-----------------|----------------------|--------------------|--------------|-------------------|---------------|----------------|------------------|----------------|
> | RVT               | 71.2 ± 6.9     | 81.6 ± 5.4      | 72.0 ± 0.0           | 88.0 ± 2.5         | 93.6 ± 4.1   | 88.0 ± 5.7        | 52 ± 2.5      | 99.2 ± 1.6     | 28.8 ± 3.9       | 48.0 ± 5.7     |
> | VidMan(multiview) | 94.4 ± 3.6     | 97.6 ± 4.4      | 92.8 ± 1.8           | 90.4 ± 2.2         | 96.8 ± 3.3   | 83.2 ± 1.8        | 88 ± 2.8      | 84.8 ± 3.5     | 48 ± 0.0         | 66.4 ± 2.1     |
> | **Models**        | **put in safe** | **place wine** | **put in cupboard** | **sort shape** | **push buttons** | **insert peg** | **stack cups** | **place cups** | **Avg. score** | **Inf. Speed(in fps)** |
> | RVT               | 91.2 ± 3.0      | 91.0 ± 5.2     | 49.6 ± 3.2          | 36.0 ± 2.5     | 100.0 ± 0.0      | 11.2 ± 3.0     | 26.4 ± 8.2     | 4.0 ± 2.5      | 62.9          | 11.6           |
> | VidMan(multiview) | 67.2 ± 3.3      | 79.6 ± 0.0     | 32.8 ± 3.3          | 48.0 ± 0.0     | 89.6 ± 1.6       | 21.6 ± 3.6     | 18.2 ± 1.8     | 13.2 ± 1.8     | 67.4   | 18.3           |
>
> It's important to mention that, given our current implementation and the video DiT backbone we've adopted, which were not originally optimized for diverse camera inputs, we anticipate further performance improvements as these methodologies evolve. Our method, as a pioneering effort, holds significant value for the field.
>
> **Q2:** `More Generated Video Results.`
>
> **A2:** To avoid violating the double-blind policy, we have provided more complex and diverse frame predictions in the attached PDF. An anonymous link to showcase our results will be provided after the rebuttal process. In particular, to effectively convey the dynamics of predicted frames, the new results are created with a sampling frequency reduced to one-third of what was originally presented in the manuscript.
>
> **Q3:** `Scalability.`
>
> **A3：** As the reviewers have noted, we have indeed observed a slight improvement with the pretrained Open-Sora model. We suspect that this could be attributed to two main
> factors: First, there may be a domain gap between the internet data utilized by Open-Sora and the robotic data we are working with. Second, to expedite the training process, we opted to load only the 12 layers from the pretrained Open-Sora, rather than the complete set of parameters. To validate these hypotheses, we conducted an experiment where we replaced our initial training data with Ego4D and fully finetune the backbone, mirroring the approach taken by GR-1. This dataset is rich in first-person perspective videos depicting hand-object interactions. We applied the same preprocessing techniques as documented in [GR-1] and [R3M]. The outcomes of our method are as follows.
>
> | **Type**   | **RLBench-10** |
> |------------|----------------|
> | w/ webdata | 51.2           |
> | w/ Ego4D   | 55.5           |
> | w/ OXE     | 57.1           |
>
> When the domain gap between the pre-training dataset and the expert-annotated robotic trajectory dataset narrows, the performance will also become closer, even if it is non-robot domain data, which supports our hypothesis. In conclusion, our method can have the similar scalability to non-robot domain data to existing methods.
>
> **Q4:** `Complete tasks.`
>
> **A4:** We apologize for the oversight regarding the results of several RLBench tasks that demand high-precision control, which were missing from Tab. 1 in the original manuscript. In our experiments, we noted that the baseline methods did not perform as strongly on these tasks as they did on others. Therefore, in our previous submission, we chose to focus on the comparisons that are more competitive to save space. After viewing the reviewers' feedback, we recognize that including the results in Tab. 1 would be less confusing for readers. Accordingly, we have revised our manuscript and incorporated the missing results as suggested.
>
> | **tasks** | **insert peg** | **stack cups** | **place cups** |
> |-----------|----------------|----------------|----------------|
> | PerAct    | 0              | 0              | 0              |
> | ori       | 12             | 8              | 4              |
> | multiview | 21.6           | 18.2           | 13.2           |

---

> ### Author Response · Authors · 2024-08-07
> **Response to Reviewer yiak [3/3]**
>
> **Q5:** `Accuracy metric.`
>
> **A5:** Following Octo, we use continuous action space. XYZ accuracy denotes did we predict the XYZ delta within 0.5 radians and 50% norm when moving. Euler angle accuracy denotes did we predict the rotations delta within 0.5 radians when moving. These two metrics can be found in Octo's codebase.
>
> **Q6:** `Computational efficiency.`
>
> **A6:** During inference, our method consumes 18 GB on a single NVIDIA 4090 GPU. We used the action chunk method with a chunk size of 5, achieving a speed of 18.3 fps.
>
> References:
>
> [GR-1]Unleashing Large-Scale Video Generative Pre-training for Visual Robot Manipulation.
>
> [R3M] R3M: A Universal Visual Representation for Robot Manipulation.

---

> > ### Comment · Reviewer_yiak · 2024-08-09
> >
> > Dear Authors,
> >
> > Thank you for your comprehensive rebuttal. Your response has thoroughly addressed my concerns.
> >
> > **Comparison with RVT and Multi-View Adaptation:** I appreciate the inclusion of comparisons with RVT and the adaptation of your model for multi-view inputs. This additional context gives me a better idea of your model's capabilities and performance relative to prior work.
> >
> > **New SIMPLER and CALVIN Results:** The added results on the SIMPLER and CALVIN benchmarks are valuable, and I am impressed that they have been implemented on such short notice. These results further demonstrate the performance of your approach across different evaluation settings. While the inclusion of real-world results would be ideal, the SIMPLER environment is a decent proxy for this. Thank you for including this at my request.
> >
> > **Inclusion of Missing RLBench Tasks:** The inclusion of the remaining RLBench tasks alleviates my concerns about cherrypicking and provides more comprehensive results.
> >
> > **State-of-the-Art Performance on RLBench-100 and CALVIN:** The state-of-the-art results on RLBench-100 and CALVIN are impressive and affirm the effectiveness of VidMan.
> >
> > **Scalability with Large-Scale Egocentric Data:** The new experiments showing improved performance from pre-training on large-scale egocentric data (Ego4D) mostly address concerns about scalability and the effective utilization of non-robot data.
> >
> > **Video Prediction Results:** The additional video prediction results in the provided PDF are compelling and provide a clearer demonstration of your video prediction model's capabilities.
> >
> > These results show that VidMan consistently outperforms several well-known works (RVT, PerAct, Octo, GR-1, Act3D) on several competitive benchmarks (CALVIN, RLBench, OXE), and it appears that the performance is definitively state-of-the-art on CALVIN ABC->D (http://calvin.cs.uni-freiburg.de/) and nearly state-of-the-art on RLBench (https://paperswithcode.com/sota/robot-manipulation-on-rlbench).
> >
> > Given the state-of-the-art performance on several baselines, the insight of your novel two-stage training approach, real-time inference ability (18.3 Hz), extremely thorough experiments and ablations, and thorough response, I have updated my rating from a 7 to a 9. **I believe this work is award-quality and will continue to advocate for it despite mixed reviews.**

---

> > > ### Author Response · Authors · 2024-08-09
> > >
> > > Dear Reviewer,
> > >
> > > We sincerely appreciate your thoughtful and detailed feedback. We are delighted that the additional experiments and comparisons we provided have addressed your concerns and that you find our work to be of high quality.
> > >
> > > Your suggestions have significantly contributed to the improvement of our paper, and we are grateful for your recognition of our efforts to include comparisons, additional benchmarks, and scalability experiments. We are especially pleased that you consider our work to be at the state-of-the-art level in multiple aspects and appreciate your support and advocacy for our work.
> > >
> > > We will continue to refine and improve our research in the revised version, and we look forward to furthering the impact of this work within the community.
> > >
> > > Thank you once again for your valuable input and for updating your rating. We are honored by your positive evaluation.

---

### Author Rebuttal · Authors · 2024-08-07

We extend our heartfelt thanks to the reviewers for their time, thoughtful suggestions, and invaluable feedback. We are honored by the positive recognition from the reviewers regarding the technical contribution (all reviewers), method insight (Reviewer yiak, 11a6, NqNP), thorough ablation (Revewer yiak, 11a6) comprehensive experiments (Reviewer yiak) and good presentation (all reviewers) and results (Revewer yiak, 11a6, 5fDE) and generalization (Reviewer yiak).

Addressing the comments, beyond the minor conceptual misunderstandings that can be readily resolved with further clarification, we identified that most concerns revolve around the following key points, and we have addressed them properly:

1. **Comparison with State-of-the-Art (SOTA) Methods**:
   - Particularly, the absence of comparisons with GR-1 on the CALVIN benchmark and RVT on RLBench was noted by Reviewers yiak, 11a6, NqNP, and Yu5P.
   - To address this, we have introduced new comparisons with GR-1 on the CALVIN benchmark and RVT, ACT3D on RLBench (Tab.1 in appended pdf file). Additionally, for methods trained with OXE, we have included comparisons on the SIMPLER environments (Tab.3 in appended pdf file). Excitingly, our method continues to demonstrate superior performance across all these scenarios, reinforcing the strength of our comparative analysis.


2. **Scalability to Non-Robot Data**:
   - Reviewers yiak, NqNP, and Yu5P questioned the method's scalability to leverage non-robotic data.
   - We have conducted experiments pretraining VidMan with Ego4D data, which consists of human manipulation videos, and observed a more significant performance improvement compared to web data (Tab.2 in appended pdf file). The reason Ego4D outperforms the web data from open-sora is likely due to the latter's inclusion of a substantial amount of landscape and scripted videos. In contrast, Ego4D shares many operational similarities with robotic data, such as object manipulation and camera movement. One can consider the Ego4D data are collected by a humanoid robot with a human-level control policy, and therefore, these non-robot data also fit into our category of robotic domain data.


3. **Compared to Other Methods's Motivation**:
   - Reviewer NqNP inquired about the distinction between our method and diffusion-diffuser, while Reviewer 5fDE questioned the difference from representation learning methods like R3M.
   - The diffusion-diffuser method generates a sequence of future states explicitly for action prediction, whereas our method avoids this explicit prediction process, leading to more efficient and real-time action predictions. Additionally, our design is specifically tailored to leverage extensive robotic video data, featuring a two-stage training approach and a layer-wise adapter. R3M employs direct supervision on the latent representation, utilizing contrastive learning and other optimization techniques, and subsequently uses this enhanced representation for action prediction after fine-tuning. In contrast, our approach aims to enhance the effectiveness of the modified cross-attention mechanism by training the Open-Sora layers and aggregating more valuable information through the fine-tuned adapter, thereby reducing loss.




We have provided detailed responses to each of the reviewers' questions and hope that our rebuttal and revisions will thoroughly address their concerns.

---

> ### Comment · Reviewer_yiak · 2024-08-07
>
> My response to the rebuttal was previously written here, but I have moved my (updated) response as a reply to the individual rebuttal now that individual rebuttals are visible to all reviewers.

---

### Decision · Program_Chairs · 2024-09-25

**Decision:**

Accept (poster)

**Comment:**

The submission initially received mixed reviews; the authors did a great job during the rebuttal, after which all reviewers became positive about the submission.  The AC agrees with the recommendations.  The authors should incorporate the rebuttal into the camera ready.